


**Clumped isotopes in near surface atmospheric CO$_2$ over land, coast and ocean in**

2                              **Taiwan and its vicinity**

3                    Amzad H. Laskar[1] and Mao-Chang Liang[1,2,3,4]*

[1]Research Center for Environmental Changes, Academia Sinica, Taipei, Taiwan
[2]Graduate Institute of Astronomy, National Central University, Taiwan
[3]Institute of Astronomy and Astrophysics, Academia Sinica, Taiwan
[4]Department of Physics, University of Houston, USA
*To whom correspondence should be addressed:      mcl@rcec.sinica.edu.tw

11                              Phone: (02) 2653-9885 #852

12                              Fax:     (02) 2783-3584





**Abstract**
Molecules containing two rare isotopes (e.g., $^{13}C^{18}O^{16}O$ in $CO_2$), called clumped isotopes, are
powerful tools to provide an alternative way to independently constrain the sources of $CO_2$ in
the atmosphere because of their unique physical and chemical properties. We present
clumped isotope data ($\Delta_{47}$) in near surface atmospheric $CO_2$ from urban, sub-urban, ocean,
coast, high mountain (~3.3 km a.s.l.) and forest in Taiwan and its vicinity. The primary goal
of the study is to use the unique $\Delta_{47}$ signature in air $CO_2$ to show the extents of its deviations
from thermodynamic equilibrium due to different processes in a variety of environments,
which the commonly used tracers such as $\delta^{13}C$ and $\delta^{18}O$ cannot provide. We also explore the
potential of $\Delta_{47}$ in air $CO_2$ to identify/quantify the contribution from various sources.
Atmospheric $CO_2$ over ocean is found to be in thermodynamic equilibrium with the
surrounding surface sea water. Also respired $CO_2$ is in close thermodynamic equilibrium at
ambient air temperature. In contrast, photosynthetic activity results in significant deviation in
$\Delta_{47}$ values from that expected thermodynamically demonstrated using $CO_2$ collected from a
controlled greenhouse. The disequilibrium could be a consequence of kinetic effects
associated with the diffusion of $CO_2$ in and out of the leaf stomata. We also observe that $\delta^{18}O$
and $\Delta_{47}$ behave differently in response to photosynthesis unlike simple water-$CO_2$ exchange
where the time scale of equilibration of the two is similar. Additionally, the measured $\Delta_{47}$
values in car exhaust $CO_2$ are significantly lower than the atmospheric $CO_2$ but higher than
that expected at the combustion temperature. In urban and sub-urban regions, the $\Delta_{47}$ values
are found to be lower than the thermodynamic equilibrium values at the ambient temperature,
suggesting contributions from local combustion emissions.





Keywords: clumped isotopes; atmospheric $CO_2$; thermodynamic equilibrium; anthropogenic;
car exhaust



## 1. Introduction

The budget of atmospheric $CO_2$ is widely studied using the temporal and spatial variations of the concentration and bulk isotopic compositions ($\delta^{13}C$ and $\delta^{18}O$) of $CO_2$ (Francey and Tans, 1987; Francey et al., 1995; Yakir and Wang, 1996; Ciais et al., 1995a,b, 1997; Peylin et al., 1999; Cuntz et al., 2003; Drake et al., 2011; Welp et al., 2011; Affek and Yakir., 2014). $\delta^{13}C$ is useful to differentiate the exchange of $CO_2$ with the ocean and land biospheres as the photosynthetic discrimination against $^{13}C$ during exchange with land plants is higher than that associated with the chemical dissolution of $CO_2$ in the ocean (e.g., Tans et al., 1993; Ciais et al., 1995a; Francey et al., 1995; Ito, 2003; Bowling et al., 2014). $\delta^{18}O$ is used for partitioning global-scale net $CO_2$ terrestrial fluxes between photosynthesis and respiration (Francey and Tans, 1987; Farquhar and Lioyd, 1993; Yakir and Wang, 1996; Ciais et al., 1997; Peylin et al., 1999; Murayama et al., 2010; Welp et al., 2011). This is because oxygen isotopes in $CO_2$ exchanges readily with water and hence the values of $\delta^{18}O$ are different when exchanging with soil water or relatively enriched leaf water; the enrichment in $^{18}O$ in the leaf water occurs during evapotranspiration. The major limitation of $\delta^{13}C$ is that it cannot distinguish between $CO_2$ produced from high temperature combustion and low temperature respiration. $\delta^{18}O$ in atmospheric $CO_2$ is mainly controlled by various water reservoirs (ocean, leaf, and soil). In urban locations, a significant fraction of $CO_2$ may have combustion origin possessing $\delta^{18}O$ signature of atmospheric $O_2$ (Kroopnick and Craig, 1972; Ciais et al., 1997; Yakir and Wang, 1996; Barkan and Luz, 2012). The $\delta^{18}O$ values from these processes and interactions are different. As a result, $\delta^{18}O$ in atmospheric $CO_2$ has been widely used for constraining the budget of $CO_2$ (Francey and Tans, 1987; Ciais et al., 1997; Gillon and Yakir, 2001; Cuntz et al., 2003; Welp et al., 2011). However, due to its short turnover time in the atmosphere, mainly affected by presence of enzyme carbonic anhydrase in plants, soils, and surface ocean, the definite determination of the associated fluxes in $CO_2$ biogeochemical models remains inconclusive. The presence of diverse $\delta^{18}O$ reservoirs and processes such as evapotranspiration also complicates the interpretation.

The doubly substituted isotopologues or clumped isotopes such as $^{13}C^{18}O^{16}O$ in $CO_2$, whose excess over the stochastic isotopic distribution, denoted by $\Delta_{47}$, provides an additional and independent constraint to study the atmospheric $CO_2$ budget and mechanisms for $CO_2$ production and consumption. Unlike bulk isotopes, clumped isotope studies for the



atmospheric $CO_2$ are very limited mainly because of the challenges to acquire it precisely
(Eiler and Schauble, 2004; Affek et al., 2007; Yeung et al., 2009). The available data are not
sufficient to address some key issues such as quantification of $CO_2$ from different sources and
to what extent the air $CO_2$ is in thermodynamic equilibrium with leaf and surface waters,
especially in regions with strong anthropogenic activities such as urban areas. Also the effect
of photosynthesis on the $\Delta_{47}$ of air $CO_2$ has not been studied rigorously. $\delta^{18}O$ and $\Delta_{47}$ were
reported to have similar isotope exchange time scales with pure water (Affek, 2013; Clog et
al., 2015), but how they behave in presence of other processes such as photosynthesis and
respiration has not been studied well. A combined assessment from all of the three
aforementioned isotopic tracers can better constrain the budget of $CO_2$ and associated
processes than $\delta^{13}C$ or $\delta^{18}O$ alone.

Theoretically it is shown that in thermodynamic equilibrium, $\Delta_{47}$ values of $CO_2$ are
temperature dependent (Eiler and Schauble, 2004; Wang et al., 2004), verified over a wide
range from 10 to 1000 $^{o}C$ (Dennis et al., 2011). Processes that involve $CO_2$ and liquid water
as medium, such as isotopic exchange with ocean water are expected to have $\Delta_{47}$ values close
to the thermodynamic equilibrium. $\Delta_{47}$ values in ambient air $CO_2$ should reflect a balance of
$CO_2$ fluxes between biosphere-atmosphere exchange, ocean-atmosphere exchange, and
emissions from combustion sources. Photosynthesis involves gas phase diffusion of $CO_2$ into
leaves, fixes ~1/3 of the $CO_2$, and returns the rest back to the atmosphere. $CO_2$ molecules
inside a leaf are generally expected to be in thermodynamic equilibrium with leaf water
because of presence of enzymatic carbonic anhydrase that greatly enhances the isotopic
exchange (Cernusak et al., 2004). $\Delta_{47}$ values of soil respired $CO_2$ is also not well constrained,
though it is believed to be in thermodynamic equilibrium with the soil water.
Here, we present clumped and bulk isotope data in near surface air $CO_2$ covering a wide
variety of processes and interactions. Air samplings were made in South China Sea, two
coastal stations in northern Taiwan, an urban traffic street, a sub-urban location, a forest site,
a greenhouse, top of a high mountain, and car exhausts. The study is designed and aimed to
show the extents of the deviations of near surface atmospheric $CO_2$ from thermodynamic
equilibrium with local surface water. Possible influences from other processes such as
anthropogenic emission, respiration, and photosynthesis on clumped isotopes are explored.
We show that $CO_2$ respired from root and soil is in close thermodynamic equilibrium with the



soil waters but photosynthesis tends to deviate it. Therefore, utilizing $\Delta_{47}$ for partitioning
fluxes between photosynthesis and respiration/soil invasion is possible.


**2. Materials and methods**
Stable isotopic compositions of $CO_2$ including mass 47 amu were measured using a Finnigan
MAT 253 gas source stable isotope ratio mass spectrometer configured to measure ion beams
corresponding to M/Z 44 through 49. The instrument registers the major ion beams (44, 45
and 46) through resistors $10^8$, $3\times10^{10}$, and $10^{11}$ Ohm, respectively, and minor ion beams (47,
48 and 49) through $10^{12}$ Ohm. All the measurements were carried out at Research Center for
Environmental Changes, Academia Sinica, Taiwan.

Air samples were collected in 2L flasks and compressed to 2 atmosphere pressure using a
membrane pump; the flasks were first flushed with the ambient air for ~10 min before sample
collection. The air was pumped through a column packed with magnesium perchlorates to
remove moisture. The moisture content was reduced from the ambient value of 70-90 % to
less than 1 % relative humidity, checked using a LI-COR infrared gas analyzer (model 840A,
LI-COR, USA).

To show how photosynthesis and respiration affect the abundances of $CO_2$ isotopologues and
to demonstrate what different information the $\Delta_{47}$ can give from the other isotopologues, we
performed systematic analyses for $CO_2$ collected in a controlled greenhouse with cemented
floor located in the top (3$^{rd}$) floor of the Greenhouse Building, Academia Sinica. The size of
the greenhouse was about 8m long, 5m wide and 5m high, and was in a condition to have
minimal air exchange with the surroundings by switching off the ventilation system. More
than 70 % of the ground area inside the greenhouse was occupied with *Cinnamomum cassia*
plants, each of ~2 m height kept in pots. Samples were collected at intervals of less than half
an hour to a few hours on three sunny days and one cloudy day to investigate the influence of
photosynthesis and respiration on the isotopologues of $CO_2$. The greenhouse was isolated
from the surroundings at least a day before the sample collection; the room relative humidity
was ~50-70 % for the three sunny days and was above 90 % for the cloudy day.



Forest air $CO_2$ was collected from a dense natural forest at the west end of the Academia Sinica Campus. The samples were collected ~100 m inside the forest on a small plateau at a height of ~30 m from the ground in the slope of a hill; the dense vegetation allowed little sunlight penetrating to the surface. The relative humidity at the site was 80-90 % during the sampling days and wind speed was nearly zero due to presence of hills on three sides of the sampling spot. Marine air was collected during a cruise in the South China Sea (for the cruise track see Figure 1) and from two coastal stations: Keelung (25°09′6″ N, 121°46′22″ E) and Fuguei Cape (25°18′ N, 121°32′ E) (Figure 1). Urban air was collected at a bus stop on Roosevelt Road, a busy street in Taipei. Sub-urban air was collected from an open roof (~30 m above ground) of Institute of Earth Science Building, Academia Sinica (AS; 25°2′41″ N, 121°36′52″ E); grassland air was collected from a grass field in front of the Department of Atmospheric Science, National Taiwan University Campus (NTU; 25° 1′ N, 121°30′ E), Taipei. In addition, we collected air from the summit of the Hehuan mountain (24°8′15″ N, 121°16′32″ E, 3.3 km a.s.l.) (Figure 1) on 9th October, 2013. All air samplings were made when there was no rain to avoid direct interaction with the rainwater. Car exhausts were collected from a Mazda 3000cc TRIBUTE and a Mitsubishi 2400cc New Outlander, using evacuated 2L glass flasks from ~20 cm inside the exhaust pipes through a column of magnesium perchlorate.

$CO_2$ was extracted from air by cryogenic technique. Air in the flask was pumped through a series of five coiled traps, with the first two immersed in dry ice-ethyl alcohol slush (-77 °C) for trace moisture removal followed by three in liquid nitrogen (-196 °C). $CO_2$ was collected from the traps immersed in liquid nitrogen by repeated freeze-thaw technique at liquid nitrogen and dry ice temperatures for further removal of traces of water [see *Mahata et al.,* 2012 for details]. The air was pumped for 40-45 minutes at a controlled rate of ~90 mL/min using a mass flow controller; the pressure on the post mass flow controller was ~10 mm of Hg. No measurable isotopic fractionation caused by mass flow controller at this flow rate was observed, checked using several aliquots of air from a high volume compressed air cylinder (~40 L at 2000 psi). For car exhaust, an aliquot of exhaust air was transferred to a 60 mL





bottle and $CO_2$ was fully extracted cryogenically following the same protocol as discussed
above (but with mass flow controller step skipped).

$CO_2$ was further purified from other condensable species like $N_2O$, $CH_4$, and hydrocarbons
by means of gas chromatography (Agilent 6890N, with a 3.0 m × 0.3 cm stainless steel
column packed with PorapakQ 80/100 mesh, supplied by Supelco Analytical, Bellefonte, PA,
USA) with the column kept at -10 $^o$C. High purity helium (>99.9999 % supplied by Air
Products and Chemicals, Inc.) at 20 mL/min was used as carrier gas. $CO_2$ was eluted first,
followed forthwith by $N_2O$, and $CH_4$, hydrocarbons and traces of water came out much later.
To get an optimized condition for $CO_2$, we checked the separation of $CO_2$ from $N_2O$ with
varying proportions and at various temperatures (25 $^o$C to -20 $^o$C) and found a temperature of
-10 $^o$C at which column separated $CO_2$ from $N_2O$ perfectly (see Supporting Information). The
column was baked at 200 $^o$C for more than 2 hours prior to use. The conditioned column is
good for purifying three samples. At the end of the day, long baking (8-10 hours) was
performed. At the initial phase the working gas was taken from a high purity commercial $CO_2$
called AS-2 ($\delta^{13}C$ = -32.54 ‰ and $\delta^{18}O$ = 36.61 ‰) procured from a local supplier (Air
Products and Chemicals, Inc.). As the difference between the isotopic compositions of
samples and AS-2 was high, we later changed the reference to Oztech $CO_2$ ($\delta^{13}C$ = -3.59‰
and $\delta^{18}O$ = 24.96 ‰) (Oztech Trading Corporation, USA) from December 2014 onward. No
detectable difference in isotopic compositions including $\Delta_{47}$ was observed between the
analyses from different working references. All $\delta^{13}C$ values are expressed in VPDB scale and
$\delta^{18}O$ in VSMOW scale, unless specified otherwise. $\Delta_{47}$ is calculated following (Affek and
Eiler, 2006):
$$\Delta_{47}=\left[\frac{R^{47}}{2R^{13}R^{18}+2R^{17}R^{18}+R^{13}\left(R^{17}\right)^2}-\frac{R^{46}}{2R^{18}+2R^{13}R^{17}+\left(R^{17}\right)^2}-\frac{R^{45}}{R^{13}+2R^{17}}+1\right]\times 1000 \quad (1)$$

where $R^{13}$ and $R^{18}$ (ratios $^{13}C/^{12}C$ and $^{18}O/^{16}O$) are obtained by measuring the traditional
masses 44, 45 and 46 in the same $CO_2$ sample and $R^{17}$ is calculated assuming a mass
dependent relation with $R^{18}$ given by $R^{17}=R^{17}_{VSMOW}\left(R^{18}\big/R^{18}_{VSMOW}\right)^{\lambda}$, where exponent
$\lambda$=0.5164 is used for all $\Delta_{47}$ calculations (Affek and Eiler, 2006). The value of $\lambda$ varies
between 0.516 and 0.523 (Hoag et al., 2005; Barkan and Luz, 2012; Hoffmann et al., 2012;
Thiemens et al., 2014). The variation in $\Delta_{47}$ is less than 0.01 ‰ at 25 $^o$C when the exponent is
varied over the aforementioned range. This variation is comparable to the measurement





uncertainty and hence is not considered here; all the calculations are based on $\lambda$=0.5164. $\Delta_{47}$
is obtained by measuring $CO_2$ with respect to which the isotopes among various $CO_2$
isotopologues are distributed randomly ($\Delta_{47} \sim 0$ ‰). Practically, this limit is approached by
heating $CO_2$ at 1000 $^o$C for more than two hours (Eiler and Schauble, 2004; Affek and Eiler,
2006). Measurements were made with a stable ~12 volt signal at mass 44, with peak centring,
background scanning, and pressure-balancing before each acquisition started. Each sample
was analyzed for 10 acquisitions, 10 cycles each at an integration time of 8 s; the total
analysis time was approximately 2.5 h. Routine analysis of masses 48 and 49, in addition to
masses 44 to 47 was used to monitor the degree of possible interference of sample impurities
on the measurements of $\Delta_{47}$ (Ghosh et al., 2006).

Dependence of $\Delta_{47}$ on $\delta^{47}$ was derived by artificially varying the $\delta^{47}$ value by ~130 ‰ (Figure
S1 in Supporting Information). $\delta^{47}$ is approximately equal to the sum of $\delta^{13}C$ and $\delta^{18}O$
measured with respect to the working gas. The wide range in $\delta^{47}$ was obtained by
equilibrating AS-2 $CO_2$ with different waters covering a wide range of $\delta^{18}O$ (-106 to +22 ‰)
at two temperatures (17 and 32 $^o$C). $CO_2$ was separated from water-$CO_2$ mixture
cryogenically and purified using gas chromatography as mentioned earlier. The extracted
$CO_2$ was divided into two aliquots: one was directly analyzed in the mass spectrometer and
the other was measured after heating at 1000 $^o$C (to define scrambled/stochastic distribution)
for more than two hours. A weak dependence of $\Delta_{47}$ on $\delta^{47}$ with a slope of -0.0017‰/‰
($\Delta_{47}/\delta^{47}$) was observed. No pressure baseline correction was made considering the little
dependence of $\Delta_{47}$ on $\delta^{47}$ (He et al., 2012). The calibration curve was then applied evenly to
all samples to remove the dependence of $\Delta_{47}$ on $\delta^{47}$ (Ghosh et al., 2006; Huntington et al.,
2009; Dennis et al., 2011). Details are provided in the Supporting Information.

The reference frame equation or empirical transfer function can then be derived from these
three temperature experiments. All the $\Delta_{47}$ values are expressed in absolute reference frame
(ARF) (Dennis et al., 2011). The empirical transfer function for the present case is
$\Delta_{47-RF} = 1.0996\,\Delta_{47-[EGvsWG]o} + 0.9145$ with $R^2$= 0.9999 (n=3), where $\Delta_{47-RF}$ is the $\Delta_{47}$ value in
the ARF and $\Delta_{47\text{-}[EGvsWG]o}$ is the intercept of the $\Delta_{47}$ versus $\delta^{47}$ plot. To obtain the temperature
from the $\Delta_{47}$ values, we used the following relation (Dennis et al., 2011):





$$\Delta_{47} = 0.003\left(\frac{1000}{T}\right)^4 - 0.0438\left(\frac{1000}{T}\right)^3 + 0.2553\left(\frac{1000}{T}\right)^2 - 0.2195\left(\frac{1000}{T}\right) + 0.0616 \qquad (2)$$

The reproducibility (1-$\sigma$ standard deviation) for air $CO_2$ measurements was established from
three aliquots of $CO_2$ extracted from a compressed air cylinder with $CO_2$ concentration
($[CO_2]$) of ~388 ppmv. The 1-$\sigma$ standard deviations were 0.07, 0.08, and 0.01 ‰ for $\delta^{13}C$,
$\delta^{18}O$, and $\Delta_{47}$, respectively (Table S1). We also used IAEA NBS-19 carbonate standard to
check the reproducibility of our measurements routinely. For carbonate analysis, $CO_2$ was
produced by reacting with ~104 % orthophosphoric acid at 25 °C. The measured isotopic data
including $\Delta_{47}$ for NBS-19 are presented in Table 1, and the long term reproducibility is 0.014
‰ (1-$\sigma$ standard deviation; n=15). The accuracy from the measurements of NBS-19 is
difficult to check, due to poor consensus of the reported $\Delta_{47}$ values from different
laboratories; our values fall within the range. To further verify the accuracy, we equilibrated
cylinder $CO_2$ (AS-2) with water at 15±2 °C and 25±2 °C, chosen to represent the ambient
temperatures presented in the current study. The deviation of temperature from the expected
values obtained from $\Delta_{47}$ was found to vary between -1 to +3 °C (Table S2).
For $[CO_2]$ measurements, flasks of volume 350 cc were used. These small flasks were
connected in series with the larger flasks used for isotopic measurements. $[CO_2]$ was
measured using a LI-COR infrared gas analyzer (model 840A, LI-COR, USA) at 4 Hz,
smoothed with 20-s moving average. The analyzer was calibrated against a working standard
(air compressed in a cylinder) with a nominal $[CO_2]$ of 387.7 ppmv and a $CO_2$ free $N_2$
cylinder. The reproducibility of LI-COR is better than 1 ppmv. The working standard was
calibrated using a commercial Picarro analyzer (model G1301, Picarro, USA) by a series of
NOAA/GMD certified tertiary standards with $[CO_2]$ of 369.9, 392.0, 409.2, and 516.3 ppmv,
with a precision (1-$\sigma$ standard deviation) of 0.2 ppmv. The $[CO_2]$ in car exhausts were
estimated by gravimetric technique using an MKS Baratron gauge.

Ambient temperatures were taken from the nearest governmental weather stations (operated
by Central Weather Bureau, Taiwan): Nankang (for AS; station code: C0A9G0; 25°03′27″
N, 121°35′41″ E, 42 m a.s.l.), Taipei (for NTU; station code: C1A730; 25°00′ 58″ N,
121°31′ 53″ E; 22 m a.s.l.), Hehuan mountain (station code: C0H9C1; 24°08′41″ N, 121°15′





51″ E, 3240 m a.s.l.), and Keelung coast (for the two coastal sites; station code: 466940;
25°08′05″ N, 121°43′56″ E, 26.7 m a.s.l.).

**3. Results**
**3.1 Greenhouse $CO_2$**
Intraday variation in the concentration and isotopic compositions of $CO_2$ inside the controlled
greenhouse is shown in Figure 2. The lowest [$CO_2$] and highest $\delta^{13}C$ and $\delta^{18}O$ values are
observed during late morning hours while highest [$CO_2$] and lowest $\delta^{13}C$ and $\delta^{18}O$ values are
observed during night time and early morning before sunrise (Table 2 and Figure 2A-2C),
indicating that respiration and photosynthesis play the major role in controlling the variations
of the [$CO_2$] and isotopic compositions. Keeling graphical analysis for $\delta^{13}C$ gives an intercept
of -26.32±0.40 ‰ (Figure 2D), a value expected for $C_3$ plant respired $CO_2$. The Keeling plot
for $\delta^{18}O$ gives an intercept of 30.68±0.73 ‰ (Figure 2E), which could be explained by a
combined effect of respired $CO_2$ equilibrated with soil water and kinetic fractionation
associated with the diffusion of $CO_2$ from soil to the air. The tight correlations among [$CO_2$],
$\delta^{13}C$ and $\delta^{18}O$ (Figure 2D-2F), however, suggest that photosynthesis/respiration are the
dominant processes controlling their variations and the mixing with ambient air and
anthropogenic contribution of $CO_2$ are insignificant.
In contrast, $\Delta_{47}$ shows different patterns of diurnal variability. Figures 3A-3D detail diurnal
variations in $\Delta_{47}$ in the greenhouse $CO_2$ in four different days. The first three are bright sunny
days while the last one is a dark cloudy day; to further reduce photosynthetic activity, two
layers of black cloths that cut down incident sunlight by ~50% are deployed for the last. The
measured $\Delta_{47}$ values are also compared with the thermodynamic equilibrium. The maximum
value of $\Delta_{47}$ is observed in the morning before ~8 AM and at night: the values are similar to
that expected at the ambient temperatures, indicating that the respired $CO_2$ is in close
thermodynamic equilibrium. The daytime $\Delta_{47}$ values are, in general, higher than the
thermodynamic equilibrium values. By comparing the $\Delta_{47}$ values acquired in the sunny days
with that in the cloudy day, we notice that when photosynthesis is weak, the $\Delta_{47}$ value is close
to the thermodynamic equilibrium (Figure 4). No correlation ($R^2 < 0.1$) is observed between
$\Delta_{47}$ and [$CO_2$], $\delta^{13}C$ or $\delta^{18}O$ (Figure 3A-C) except when the photosynthesis is weak (Figure





3D), which suggests that the $\Delta_{47}$ carries information different from concentration and bulk
isotopes when photosynthesis occurs. See Section 4.1 for detailed discussion.

### 3.2 Car exhaust

The concentration, $\delta^{13}C$ and $\delta^{18}O$ values of car exhaust $CO_2$ are 39350±50 ppmv, -
27.70±0.03 ‰ and 25.35±0.07 ‰, respectively (Table 3). $\delta^{13}C$ value is similar to that
reported elsewhere (Newman et al., 2008; Popa et al., 2014), the $\delta^{18}O$ is slightly higher than
the atmospheric $O_2$ (~23.5 ‰), the source of $O_2$ for combustion. This is probably due to post
isotopic exchange with water present in the stream of the exhaust inside the catalytic
converter and the exhaust pipe. The average value of $\Delta_{47}$ for the exhausts from the two cars is
0.273±0.021 ‰, which gives an average temperature of 282±17 $^{o}C$ (Table 3).

### 3.3 $CO_2$ over ocean, coasts and land

Isotopic compositions including $\Delta_{47}$ values obtained for $CO_2$ over ocean, coasts, sub-urban,
and grassland are summarized in Table 4 and 5. The averaged [$CO_2$] over ocean between
latitudes $18^{o}03'$ N and $21^{o}17'$ N is 395±7 ppmv, and the values of $\delta^{13}C$ and $\delta^{18}O$ are -
8.43±0.19 ‰ and 40.72±0.20 ‰, respectively (Table 4). Figure 5 shows a comparison of
carbon Keeling analyses for the atmospheric $CO_2$ collected over different regions. The
intercept for oceanic $CO_2$ is -15.96±1.95 ‰ (Figure 5A). In the coastal stations, the averaged
values of [$CO_2$], $\delta^{13}C$, and $\delta^{18}O$ are 397±10 ppmv, -8.48±0.11 ‰, and 40.70±0.29 ‰,
respectively, with a $\delta^{13}C$ Keeling intercept of -12.20±1.11 ‰ (Figure 5B). Both the [$CO_2$]
and $\delta^{13}C$ values over the ocean and coasts are similar to those observed at Mauna Loa during
the sampling period, suggesting little contribution from local/regional anthropogenic sources.
However, the intercepts of the Keeling plots is different from the $\delta^{13}C$ value of the $CO_2$
released by the remineralization of organic matter (-20 to -30 ‰) in the deep sea regions, the
expected source of $CO_2$ over ocean. This is probably due to partial isotopic equilibration of
the $CO_2$ with dissolved inorganic carbon before releasing to the atmosphere (see discussion
for details).



The averaged values of $[CO_2]$, $\delta^{13}C$, and $\delta^{18}O$ for air $CO_2$ near Roosevelt Road, a busy street
in downtown Taipei, are $500\pm50$ ppmv, $-11.05\pm0.90$ ‰, and $39.319\pm0.94$ ‰, respectively
(Table 5). Both the $[CO_2]$ and isotopic compositions show signatures of a significant
contribution from vehicular emissions. In the sub-urban location (AS), $[CO_2]$ averaged over
four months is $410\pm10$ ppmv (Table 5), ~15 ppmv higher than that observed over the South
China Sea and that at Mauna Loa Observatory during the time of sampling. The higher $[CO_2]$
suggests contribution from local anthropogenic emissions. $\delta^{13}C$ values mainly vary between -
7.83 to -10.30 ‰, with an average of $-8.78\pm0.50$ ‰. Keeling analysis for $\delta^{13}C$ (Figure 5C)
gives an intercept of $-26.16\pm1.58$ ‰, indicating source of $CO_2$ from $C_3$ plant respiration
and/or combustion. The averaged $[CO_2]$ over the grassland (NTU) is $410\pm33$ ppmv. The
Keeling plot intercept is $-16.98\pm1.02$ ‰ (Figure 5D), indicating a significant fraction of $CO_2$
originated from $C_4$ vegetation. This is not surprising as the $CO_2$ was sampled over a $C_4$
dominated grassland (area: ~50 m x 50 m). We note that though the station is located in an
urban region, the sampling location is at least ~150 m away from traffic streets, such as
Keelung road, along with ~60 m wide, ~10 m high $C_3$ trees in between. As a result,
anthropogenic signals are not very prominent. The averaged values of $\delta^{13}C$ and $\delta^{18}O$ are -
$8.95\pm0.70$ ‰ and $39.74\pm1.00$ ‰, respectively. Unlike greenhouse $CO_2$, no statistically
significant correlation between $\delta^{18}O$ and $1/[CO_2]$ in air $CO_2$ in these sites is observed (not
shown), probably due to various contributions from multiple sources and processes affecting
$CO_2$.
The $[CO_2]$, $\delta^{13}C$, and $\delta^{18}O$ values for two high mountain air $CO_2$ samples collected on 9th
October, 2013 are 364 ppmv, $-8.23\pm0.02$ ‰ and $40.59\pm0.30$ ‰, respectively (Table 5). The
lower $[CO_2]$ and higher $\delta^{13}C$ than Mauna Loa suggests photosynthetic uptake, which is also
seen at NTU site and inside greenhouse on a few occasions. The air $[CO_2]$, $\delta^{13}C$ and $\delta^{18}O$ are
$438\pm16$ ppmv, $-9.99\pm0.50$ ‰ and $40.39\pm0.63$ ‰, respectively, for a dense forest site near the
Academia Sinica (AS) Campus. Given the proximity of the site from AS, the higher
concentration and lower $\delta^{13}C$ values than those at AS indicate significant influence from local
respiration (Table 5).
Figure 6 shows the time series of $\delta^{13}C$ and $\delta^{18}O$ at the sub-urban station where measurements
were carried out for more than four months. Tentatively, $[CO_2]$ level increases and $\delta^{13}C$
depletes from October to February (Figure 6A), likely a result of seasonal variation in
photosynthesis/respiration. On average, the $\delta^{13}C$ value is slightly less than the global mean,



implying influence from local/regional anthropogenic activities though the dominant role is
played by biogeochemistry in affecting the variation. The time series of $\delta^{18}O$ (Figure 6B)
shows variation between 39.40 and 41.57 ‰, with an average of 40.87±0.46 ‰. An
increasing trend is also observed in $\delta^{18}O$ from October to February. We attribute this to
interactions with rain and surface waters which are heavier in winter time compared to the
summer (Peng et al., 2010; Laskar et al., 2014).
The $\Delta_{47}$ values vary between 0.880 ‰ to 0.946 ‰ for the marine and coastal $CO_2$ (Table 4,
Figures 7A and 7B), similar to that predicted at thermodynamic equilibrium at sea surface
temperatures (obtained using equation (2)). Similarly, $\delta^{18}O$ of air $CO_2$ shows the expected
equilibrium values with the surface sea water (see discussion), suggesting that the air $CO_2$ is
indeed in thermodynamic equilibrium with the underlying sea water. Figure 7C shows the
measured $\Delta_{47}$ values at the sub-urban station along with the equilibrium values expected at
the ambient temperatures. Here the $\Delta_{47}$ values vary between 0.853 ‰ and 0.972 ‰, which in
contrast to the marine $CO_2$, are significantly less than the thermodynamic equilibrium values
(assuming water bodies have the same temperature as the ambient) (Table 5). Figure 7D
shows the $\Delta_{47}$ values in the grassland (NTU). A large variation in $\Delta_{47}$ is observed (0.885 -
0.989 ‰) with an average of 0.937±0.030 ‰; some of the values are close to the
thermodynamic equilibrium while the others deviated significantly. The forest air $CO_2$ $\Delta_{47}$
values in summer fall in the range of 0.887 ‰ to 0.920 ‰, with an average of 0.895±0.012
‰ (Table 5). The values are similar to that at thermodynamic equilibrium (Figure 7E) except
on 11[th] August, when a significant increase in $\Delta_{47}$ was observed. The deviation is probably
due to influence of a super typhoon, which passed over the region on previous days mixing
and transporting air masses regionally. In the high mountain station, the averaged value of
$\Delta_{47}$ is 0.904±0.009 ‰, slightly less than that expected at the ambient temperature (Table 5).
To show how anthropogenic emission affects the background $\Delta_{47}$ values, we collected several
air $CO_2$ samples from Roosevelt Road and the values are in the range of 0.754‰ to 0.833 ‰,
with an average of 0.807±0.028 ‰ (Figure 7F). The value is lower by ~0.16 ‰ compared to
the thermodynamic equilibrium value, indicating a significant fraction of $CO_2$ produced at
higher temperatures.

**4. Discussion**



As stated earlier, the $\Delta_{47}$ has the unique physical property of representing the formation
temperature of a $CO_2$ molecule, providing an alternative tool for constraining the budget of
$CO_2$ in the atmosphere. We present in detail the data of multiple $CO_2$ isotopologues obtained
from a controlled greenhouse, where atmospheric mixing and transport are largely reduced, to
demonstrate the advantage of utilizing $\Delta_{47}$ for flux partitioning between photosynthesis and
respiration over other $CO_2$ isotopologues. The data collected from other natural environments
are also presented, compared, and discussed.
In urban and industrial places where anthropogenic emission is significant, all the three
isotopic tracers, viz., $\delta^{13}C$, $\delta^{18}O$, and $\Delta_{47}$, provide information about the anthropogenic
fraction of $CO_2$ due to distinct values of their sources. For example in a traffic street, a two
end member (background and anthropogenic $CO_2$) mixing of any of these tracers may
provide sufficiently good estimate of the anthropogenic fraction of $CO_2$. However, if a
significant fraction of $CO_2$ is respired from soil under $C_3$ plants, $\delta^{13}C$ cannot distinguish
between the respired and anthropogenic sources. $\delta^{18}O$ is always not applicable due to
complexity of multiple oxygen-containing sources. Anthropogenic $CO_2$ can also be
quantified using radiocarbon ($^{14}C$) as fossil fuels are highly depleted in $^{14}C$ (Miller et al.,
2012); however, it cannot distinguish difference between $CO_2$ from two sources with modern
carbon.
The un-catalyzed isotopic exchange time scale between $CO_2$ and water is similar for both
$\delta^{18}O$ and $\Delta_{47}$ (e.g., see Affek, 2013), and therefore, we expect that the two provide similar
information when $CO_2$ in air simply exchanges with water. But it is not well understood if
they behave similarly when biogeochemical processes such as photosynthesis and respiration
are involved. We note that $^{18}O$ is highly variable between reservoirs such as leaf water
affected by evapotranspiration even when temperature variation is not very large. Thus, $\Delta_{47}$
can complement $\delta^{18}O$ and $^{14}C$ data to probe the associated processes in the $CO_2$ cycling. A
detailed analysis of the results from different locations is presented below.

**4.1 Greenhouse $CO_2$**
To minimize anthropogenic alteration and air mixing/transport and to maximize the
variations of $CO_2$ isotopologues by biogeochemical processes, a controlled greenhouse





provides an ideal environment. Diurnal variation is observed in [$CO_2$], $\delta^{13}C$, $\delta^{18}O$ (Figure 2),
and $\Delta_{47}$ (Figure 3) in the greenhouse. Good correlations between [$CO_2$], $\delta^{13}C$ and $\delta^{18}O$
suggest common processes affecting all of them, and we believe they are photosynthesis and
respiration. Giving July 31$^{st}$ as an example, we estimate the rates of night-time respiration
and daytime photosynthetic uptake using the bulk isotopic compositions ($\Delta_{47}$ which will be
discussed separately). The dimension of the greenhouse room is 8m, 5m and 5m (length,
width and height). The night-time respiration rate is then estimated to be about ~10 ppmv per
hour (considering change of [$CO_2$] from 5:30 PM to 9:30 PM; Figure 2A), or ~$4\times10^{13}$
molecules cm$^{-2}$ s$^{-1}$. The increase of [$CO_2$] can be satisfactorily explained assuming $C_3$
respiration as the main source of $CO_2$ ($\delta^{13}C \approx$ -26 ‰; intercept in Figure 2D) added to the
background (-8.5 ‰). Similarly, the same conclusion is also arrived by analyzing $\delta^{18}O$ (the
respired $CO_2$ is 30.68 ‰, intercept in Figure 2E, and background, $\delta^{18}O$ of air $CO_2$ outside, is
40 ‰). Thus, we conclude that the main factor that affects the changes in concentration as
well as the isotopic compositions in night-time is respiration.
The daytime net uptake rate can be estimated by taking the changes from early morning to
noon time; the [$CO_2$] reduces by 110 ppmv, $\delta^{13}C$ increases by 3.46 ‰, and $\delta^{18}O$ by 2.23 ‰ in
about six hours. The estimated net photosynthetic uptake is ~$7\times10^{13}$ molecules cm$^{-2}$ s$^{-1}$.
Neglecting respiration during daytime, the photosynthetic discrimination can be calculated
using the Rayleigh distillation model

$$R = R_o f^{\alpha-1} \tag{3}$$

where $R_o$ and $R$ are the initial and photosynthesis modified $^{13}C/^{12}C$ or $^{18}O/^{16}O$ ratios,
respectively, $f$ is the fraction of the material left, and $\alpha$ is the fractionation factor. The
estimated discrimination in $^{13}C$ defined by ($\alpha$-1), following equation (3), is -15.3 ‰, similar
to that expected for $C_3$ type vegetation. For $^{18}O$, in addition to photosynthetic uptake, one has
to consider an additional effect due to temperature-dependent water-$CO_2$ equilibrium
fractionation. That is, the process decreases $\delta^{18}O$ by ~0.2 ‰ for an increase of 1 $^o$C in
temperature (Brenninkmeijer et al., 1983); from morning to noon time, the temperature effect
reduces $\delta^{18}O$ by -4.4 ‰. Adding this factor to the observed change in $\delta^{18}O$ yields a
discrimination of about -27 ‰; the value becomes -9.5 ‰, if this additional temperature-
dependence is ignored. The obtained discrimination factors for $^{13}C$ and $^{18}O$ are in good





agreement with those reported previously (Farquhar et al., 1989; Flanagan et al., 1997; Cuntz
et al., 2003; Affek and Yakir, 2014).
Assuming ca. 1/3 of the $CO_2$ molecules in stomata are fixed photosynthetically, the
remaining retro-diffuse back to the atmosphere (Farquhar and Lloid, 1993), implying that the
$CO_2$-water isotopic exchange rate is ~$2 \times 10^{14}$ molecules cm$^{-2}$ s$^{-1}$, or 9 hours of oxygen isotope
exchange time for $CO_2$ in the room. As a result, we do not expect that $CO_2$ reaches complete
isotopic equilibrium with the substrate water in a few hours. $\Delta_{47}$ values in the leftover $CO_2$
can be used to check the disequilibrium. The respired $CO_2$ are found to be always in
thermodynamic equilibrium at the ambient temperature, shown by the $\Delta_{47}$ values of $CO_2$ in
the early morning and night-time (Figure 3A-3C) and that collected on a cloudy day with
suppressed photosynthetic activity (Figure 3D). The close-thermodynamic equilibrium at
reduced photosynthetic condition is also shown in Figure 4A that deviation from the expected
is small. On sunny days, the [$CO_2$], $\delta^{13}C$, and $\delta^{18}O$ values change by 50-115 ppm, 2-4 ‰, and
1.1-2.2 ‰, respectively, in a time period of ~5 hours in the morning (Figure 2). Figure 3
shows that the $\Delta_{47}$ values retain the thermodynamic equilibrium values in the morning hours
(until 9 AM) and deviate later on. The reduction and deviation in the $\Delta_{47}$ values during the
time period is ~0.05 ‰ (Figures 3A-3C); the changes we believe are significant, as the values
are much higher than the uncertainty of the measurements. We attribute this deviation to
photosynthesis as it is seen when photosynthesis is strong. Strong influence of photosynthesis
on $\Delta_{47}$ was also reported previously (Eiler and Schauble, 2004). Photosynthesis as a source of
disequilibrium was further shown recently by analyzing the clumped isotopes of $O_2$ (Yeung
et al., 2005). Though enzymatic carbonic anhydrase catalyzes the water-$CO_2$ isotopic
exchange toward equilibrium (Peltier et al., 1995; Cernusak et al., 2004), the reaction may
not complete, limited by the enzymatic activity inside leaves; large variation in the activity of
carbonic anhydrase in different vegetation types ($C_3$, $C_4$) or within the same type is observed
(see Gillon and Yakir, 2001 and references therein). Furthermore, a box modeling by Eiler
and Schauble (2004) demonstrated that gas diffusion through leaf stomata during
photosynthesis fractionates the remaining air $CO_2$ $\Delta_{47}$ values from the thermodynamic
equilibrium set by leaf water. Mixing of more than one component can also cause change in
$\Delta_{47}$ when bulk isotopic compositions of the components are different (Affek and Eiler, 2006),
but this can easily be ruled out as it is not observed when photosynthesis is not very strong
(Figure 3D). More rigorous investigations with controlled experiments using different plants



with diverse carbonic anhydrase activities are needed to resolve the issue. We note that no
significant correlation between $\delta^{18}O$ and $\Delta_{47}$ is observed (Figure 3). Therefore, the plant
photosynthesis decouples $\Delta_{47}$ and $\delta^{18}O$; in contrast, pure water-$CO_2$ isotopic exchange
process shows that the two behave similarly as far as isotopic equilibration is concerned
(Affek, 2013; Clog et al. 2015).
The $\Delta_{47}$ thus serves as an independent tracer for studying photosynthesis. Though the
deviation from equilibrium during photosynthesis is also observed in oxygen clumped
isotopes [*Yeung et al.*, 2015], $CO_2$ and $O_2$ are affected and produced from different processes
and sources; the former is affected seriously by water (water-$CO_2$ isotopic exchange) while
the latter is derived from water. We believe the analyses of the clumped isotopes for both
$CO_2$ and $O_2$ are of great importance in the atmospheric carbon cycling study, providing a new
angle for tackling the chemistry chain in photosynthesis. More systematic study in controlled
environments including leaf level experiments will help to better understand the role of
photosynthesis on $\Delta_{47}$.

## 4.2 Marine and coastal air $CO_2$

The concentration and $\delta^{13}C$ values of marine air $CO_2$ are close to the background atmospheric
values reported at Mauna Loa, indicating little contribution from local/regional anthropogenic
activities. The Keeling analysis for $\delta^{13}C$ gives an intercept of $-15.9\pm2.0$ ‰ (Figure 5A) which
is the $\delta^{13}C$ value of the source $CO_2$ over the ocean. The $CO_2$ released over ocean is mainly
originated from the remineralization of organic matter in the deeper ocean, the $\delta^{13}C$ value of
which ranges between -20 and -30 ‰ in the tropical to subtropical oceans (Francois et al.,
1993; Goericke and Fry, 1994), the intercept observed here is much higher than this range. A
possibility is that the remineralized $CO_2$ gets equilibrated with the dissolved inorganic carbon
before releasing. Again a complete equilibration of the $CO_2$ with the dissolved inorganic
carbon would lead to a $\delta^{13}C$ value of released $CO_2$ to be -9 to -10 ‰ (Mook, 1986; Boutton,
1991; Zhang et al., 1995; Affek and Yakir, 2014), the observed value of the intercept is much
less than this. Therefore, we conclude that the $CO_2$ produced in the deeper ocean is partially
equilibrated with the dissolved inorganic carbon before releasing to the atmosphere.



The $\delta^{18}$O values of the surface sea water in the region in summer (July-September) and
winter (December-February) are about -1.7 ‰ and -0.6 ‰ (Ye et al., 2014). The sea surface
temperatures in the summer and winter are about 28 and 24 $^o$C, and the equilibrated $\delta^{18}$O
values of the atmospheric $CO_2$ should be 38.9 ‰ and 40.7 ‰, respectively (Brenninkmeijer
et al., 1983). Our observed values lie in the range of 40.4 ‰ to 41.0 ‰ (Table 4), consistent
with the isotopic equilibrium values with the surface water. Therefore, we conclude that
oxygen isotopes in near surface air $CO_2$ over ocean are close to the isotopic equilibrium with
the surface sea water. This conclusion is further supported by the observed $\Delta_{47}$ values. This is
due to the same water-$CO_2$ exchange time for the two species (Affek, 2013; Clog et al.,
2015). Comparing the greenhouse data above, we therefore conclude that $\delta^{18}$O and $\Delta_{47}$
respond differently when photosynthesis is the main governing factor and behave similarly
when exchange occurs due to simple water-$CO_2$ equilibration. Though carbonic anhydrase
are also present in the surface ocean and marine phytoplankton does photosynthesis, $\delta^{18}$O and
$\Delta_{47}$ in air $CO_2$ over the ocean show the values at thermodynamic equilibrium unlike
greenhouse. The degree of deviation from thermodynamic equilibrium likely increases with
the strength of photosynthesis, and normally the oceanic photosynthesis is less compared to
the terrestrial plants. Therefore, $\Delta_{47}$ can be used as a tracer for estimating terrestrial carbon
uptake. Compared to $\delta^{18}$O, $\Delta_{47}$ is process sensitive and is not affected by the isotopic
composition of substrate water. Given that the surface air temperature is better measured, we
believe the clumped isotopes potentially provide good tracers for global carbon flux study
involving $CO_2$, complementing the commonly used species like [$CO_2$], $\delta^{13}$C, and $\delta^{18}$O.
The isotopic values including $\Delta_{47}$ in the two coastal stations are similar to those observed for
the marine $CO_2$. The carbon Keeling analysis yields an intercept of -12.20±1.11 ‰ (Figure
5D), consistent with that for the marine $\delta^{13}$C (removing one outlier from Figure 5A gives an
intercept of -13.3 ±1.0 ‰). The $\Delta_{47}$ values here are similar to the thermodynamic equilibrium
with the sea surface water at the temperature of ~27 $^o$C (Figure 7B). The recoded air
temperature during the sampling period over the coasts varies between 14 and 24 $^o$C and is
not reflected in the $\Delta_{47}$ values. We note that the samples are collected from two open spaces
in the coasts where strong north and northeasterly winds overwhelm, carrying air masses
from the oceans towards the sampling locations (See Table S3 in Supporting Information).
Therefore, we expect the major contribution is marine air with little influence from local



processes, which could occasionally cause deviation from the thermodynamic equilibrium
values.

**4.3 Car exhaust CO$_2$**
The $\Delta_{47}$ value of car exhaust CO$_2$ should reflect the temperature of fuel combustion inside the
combustion chamber which is >800 $^{\circ}$C. However, the temperature estimated from $\Delta_{47}$ is
found to be 283±18 $^{\circ}$C. It is likely that interaction of the sample CO$_2$ with the condensed
water in the exhaust pipe modifies the $\Delta_{47}$ value: during sample collection, we observed that
the exhaust gas contains a large amount of water vapor and some of which get condensed on
the exhaust pipe and the front part of the magnesium perchlorate column. Precautions, such
as opening the evacuated flask for a short time (<1 min) and careful holding of the sampling
tube inside the exhaust pipe without touching the wall of the pipe, are taken to minimize
CO$_2$-water interaction during sample collection.
The higher $\Delta_{47}$ value for the exhaust CO$_2$ indicates isotopic re-equilibration of CO$_2$ with
water in the stream of the exhaust gas and inside catalytic converter, also supported by the
observed enriched $\delta^{18}$O than atmospheric O$_2$; the oxygen atoms in the two most abundant
species, water and CO$_2$ here, are mostly originated from atmospheric O$_2$ and are expected to
inherit the isotopic composition of atmospheric O$_2$. Normally isotopes in CO$_2$ do not
exchange with water vapor, but inside catalytic converter, exchange may take place on the
surface of the catalyst at certain temperature (which is usually much less than the combustion
temperature). Affek and Eiler (2007) also observed elevated $\Delta_{47}$ values for car exhausts and
estimated a temperature of CO$_2$ production to be ~200 $^{\circ}$C. The temperature estimated here is
significantly higher than that observed by Affek and Eiler (2007). Difference could be due to
different car models and the variations in the temperatures of the catalytic converters from car
to car.

**4.4 Urban and sub-urban air CO$_2$**
A significant fraction of anthropogenic CO$_2$ is present in the air CO$_2$ over the urban site,
indicated by the [CO$_2$] as well as isotopic compositions including $\Delta_{47}$. Limits to the





anthropogenic contribution can be estimated following a two component mixing: $\delta$ =
$f_{anth} \times \delta_{anth} + (1-f_{anth}) \times \delta_{bgd}$, where $\delta$'s can be $\delta^{13}C$ or $\delta^{18}O$ or $\Delta_{47}$ and f's, the corresponding
weighting factor, and subscripts 'anth' and 'bgd' refer to anthropogenic and background,
respectively. We take the 'anthropogenic' and 'background' end member isotopic
compositions from the car exhaust (Table 3) and marine $CO_2$ (Table 4), respectively.
Assuming that the excess in $[CO_2]$ above the background is originated from vehicular
emissions, the values of the $\delta^{13}C$, $\delta^{18}O$, and $\Delta_{47}$ in the urban site obtained using the mixing
equation are -12.26 ‰, 37.68 ‰, and 0.791 ‰, respectively, which are similar to those
observed (Table 5). $\Delta_{47}$ is not a conserved quantity and a linear mixing is not valid when the
bulk isotopic compositions of the components are widely different. In the present case, the
isotopic compositions of the two components are not drastically different and fraction of
anthropogenic $CO_2$ is much less (<1/4) than the background $CO_2$, and hence the error due to
linear approximation is smaller than the uncertainty of measurement.
No systematic diurnal or temporal trend is observed in the $\Delta_{47}$ values in sub-urban $CO_2$
during the sampling period (Figure 7C) though a weak trend is seen in $\delta^{13}C$ and $\delta^{18}O$ (Figure
6), furthermore demonstrating that $\Delta_{47}$ behaves differently from $[CO_2]$, $\delta^{13}C$, and $\delta^{18}O$.
Almost all measured $\Delta_{47}$ values are lower than that expected at the ambient temperature
except two days: 9[th] November, 2013 and 3[rd] February, 2014. $\delta^{13}C$ values are also slightly
lower than the background values. The reduced values of $\Delta_{47}$ could be due to contribution of
$CO_2$ from combustion processes which produce $CO_2$ with low $\Delta_{47}$ values as discussed in
Section 4.3. We estimate the contribution of local anthropogenic emissions in $\delta^{13}C$ and $\Delta_{47}$
using the two components mixing discussed above. The components are the background air
$CO_2$ and car exhausts. The expected $\delta^{13}C$ and $\Delta_{47}$ values of the mixture are -9.1 ‰ and 0.92
‰, respectively. The observed $\Delta_{47}$ value is significantly different from that estimated from
simple two component mixing, though it is not different for $\delta^{13}C$, suggesting other processes
like photosynthesis present in affecting $\Delta_{47}$. After subtracting the local anthropogenic
contribution from the observed $\Delta_{47}$ values, a difference of ~0.026 ‰ between the observed
and estimated remains for sub-urban station and it disappears for urban station (see Table S4
in Supporting Information). This is not obvious in $\delta^{13}C$ probably due to larger variation. The
lower $\Delta_{47}$ values in sub-urban station could possibly be due to kinetic effect during
photosynthetic assimilation, partial contribution of marine air, or a combination of them. The
marine air in the vicinity of Taiwan, which is at thermodynamic equilibrium with the surface





sea water as discussed earlier, may contribute partially to the air $CO_2$ at the sampling site.
Varying contribution of marine air could explain the lower $\Delta_{47}$ values to some extent. The
respired $CO_2$ is in thermodynamic equilibrium as shown above (Section 4.1). Therefore, the
most plausible cause for observed deviation in the $\Delta_{47}$ values that cannot be accounted for by
anthropogenic and marine alterations is photosynthesis, as discussed earlier for greenhouse
$CO_2$. This is not unreasonable, as the Academia Sinica Campus is surrounded by thick
greeneries.
On $9^{th}$ Nov, 2013 and $3^{rd}$ February, 2014, the $\Delta_{47}$ values are close to that expected at
thermodynamic equilibrium (Figure 7C). The $\Delta_{47}$ values on $9^{th}$ November are not very
different from the values reported for the previous or next days. However, the calculated
thermodynamic equilibrium values on that day are relatively low due to higher ambient
temperatures (Figure 7C). On $3^{rd}$ Febrauray, 2014, the $\Delta_{47}$ values are higher and comparable
to the thermodynamic equilibrium values expected at ambient temperatures. A likely
explanation is that on that day relatively strong wind from the southern land (Table S3 in
Supporting Information) contributed the air $CO_2$ and higher $\Delta_{47}$ values are due to mixing of
the local air with that transported from the south of Taipei.

### 623    4.5 Forest, grassland and high mountain air $CO_2$

An elevated $CO_2$ concentration and low $\delta^{13}C$ and $\delta^{18}O$ values indicate significant contribution
of respiration and/or anthropogenic $CO_2$ in the forest station (Table 5) near the Academia
Sinica Campus. Though the samples are collected at 10-11 AM under bright sunlight, the
vegetation is so dense that little sunlight reached the ground. As a result, photosynthesis is
weakened at the ground level. Also poor circulation of air due to presence of high heels on
the three sides of the sampling spot makes the site nearly isolated from the surroundings. The
$\Delta_{47}$ values are similar to the thermodynamic equilibrium expected at the ambient
temperatures except on $11^{th}$ August, 2015 on which a significantly higher $\Delta_{47}$ value is
observed (Figure 7F). The higher value is likely due to the influence of the super Typhoon
Soudelor which passed over Taipei during 8-10 August, 2015 causing a decrease in
temperature by 3-4 $^{o}C$ and air masses mixing in a larger spatial scale.



In the grassland station in Taipei city, the Keeling plot for $\delta^{13}C$ gives an intercept of -17.0±1.0 ‰ (Figure 5D). This indicates some sources of $CO_2$ with higher $\delta^{13}C$ values compared to the most expected sources, namely, $C_3$ vegetation and vehicle emission with a $\delta^{13}C$ value of ~ -27 ‰. The samples are collected just above the surface of the grasses. Tropical warm grasses are mainly $C_4$ type with $\delta^{13}C$ in the range of -9 to -19 ‰ and a global average of -13 ‰ (Deines, 1980). We measured $\delta^{13}C$ values of a few grass samples and found values in the range of -15 to -17 ‰. The soil and grass respired $CO_2$ with higher $\delta^{13}C$ contributed significantly to the near surface $CO_2$, resulting in an elevated intercept of -17 ‰. The concentration is sometimes observed to be less than the background level, probably due to strong $CO_2$ uptake by plants. The temperature gradually decreased from 26 to 20 $^oC$ during the consecutive three days and clumped isotope followed similar trend, reflecting the influence of temperature on $CO_2$ $\Delta_{47}$ and rapid equilibration with the leaf and surface waters. The low value observed on the second day is probably due to plumes of vehicle exhausts, supported by the elevated level in [$CO_2$] and depletion in $\delta^{13}C$ and $\delta^{18}O$ (Table 5).

For high mountain $CO_2$, the $\Delta_{47}$ value (Table 5) is lower than that expected at ~10 $^oC$, the ambient temperature at the top of the mountain site during sampling. The $\Delta_{47}$ values are similar to that observed in the plain and over the ocean. We note that during the sampling period, the site was affected significantly by winter monsoons. HYSPLIT 24 hours back trajectory shows marine origin of air (not shown) during the sampling time. The air $CO_2$ on the mountain probably does not get sufficient time to isotopically equilibrate with the local surface and leaf water but show the signature of the marine $CO_2$.

The deviations in $\Delta_{47}$ from the thermodynamic equilibrium values in different atmospheric environments and processes are summarized in Figure 8. It is obvious that the urban and sub-urban $CO_2$ deviate the most towards lower $\Delta_{47}$ values, mainly contributed by $CO_2$ originated from high temperature combustions, i.e., vehicular emissions. The respired $CO_2$ are always in close thermodynamic equilibrium at the ambient temperature. On the other hand, $CO_2$ affected by strong photosynthesis show significant deviation from the thermodynamic equilibrium values. Kinetic isotopic fractionation during diffusion of $CO_2$ in and out of leaf stomata is a probable reason.

**5. Summary**



We presented a compilation of $\Delta_{47}$ analyses for car exhaust, greenhouse and air $CO_2$ over a wide variety of interactions in tropical and sub-tropical regions including marine, coastal, urban, sub-urban, forest, and high mountain environments. Car exhaust, urban, sub-urban and greenhouse air $CO_2$ significantly deviate from the thermodynamic equilibrium values. While respired $CO_2$ is in thermodynamic equilibrium with leaf and soil surface waters, photosynthesis significantly deviates the $\Delta_{47}$ values from the thermodynamic equilibrium. The $\Delta_{47}$ values in urban and sub-urban air $CO_2$ are lower than that expected under thermodynamic equilibrium at the ambient temperature. The deviation is mainly due to contributions from fossil fuel emissions and to some extent due to photosynthesis especially in regions with dense vegetation. We expect $\Delta_{47}$ can shed light on the estimation of anthropogenic contribution to the atmospheric $CO_2$ and the activity of photosynthesis. The latter deserves further investigation, to establish how exactly $\Delta_{47}$ is affected by photosynthesis, before the tracer can be used for estimating gross primary productivity.

**Data availability**

All the data used in the manuscript are also presented in the form of Tables.

**Acknowledgement**

We thank Dr. Chung-Ho Wang for providing waters with different $\delta^{18}O$, Institute of Earth Sciences, Academia Sinica for providing laboratory space, Mr. Frank Lin for helping sampling in greenhouse, Dr. Jia-Lin Wang and Dr. Chang-Feng Ou-Yang for calibrating compressed air cylinder, Mr. Hao-Wei Wei for collecting air at the campus of National Taiwan University and Mr. Wei-Kang Ho for collecting oceanic $CO_2$ and helping in laboratory setups. Special thanks to Prof. S. K. Bhattacharya and Dr. Sasadhar Mahata for helpful discussion. This work is supported by the Ministry of Science and Technology (MOST-Taiwan) grants 101-2628-M-001-001-MY4 and 103-2111-M-001-006 to Academia Sinica and Academia Sinica Career Development Award and MOST 103-2119-M-002-022 to National Taiwan University.



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



**Figures**

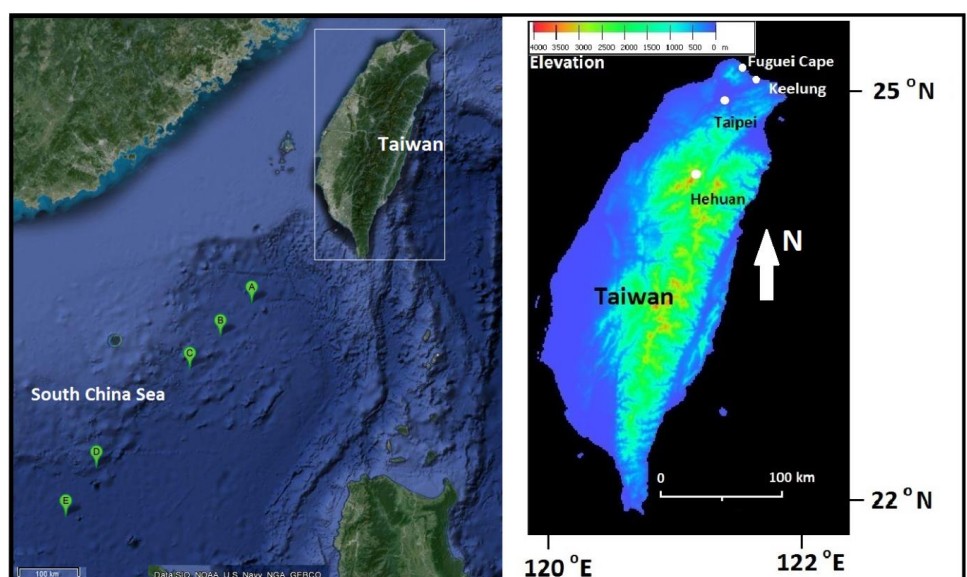


Figure 1. Map of Taiwan and South China Sea with the locations of air sampling. Marine air
$CO_2$ sampling stations (A to E) in the South China Sea are shown on the left. Fuguei Cape
and Keelung are two coastal stations, urban site (Roosevelt Road) and grassland (National
Taiwan University Campus) are located at the centre of Taipei City and sub-urban site
(Academia Sinica Campus) at the outskirt of the city and Hehuan is a high mountain station
(~3000 m a.s.l.); all are shown on the right.






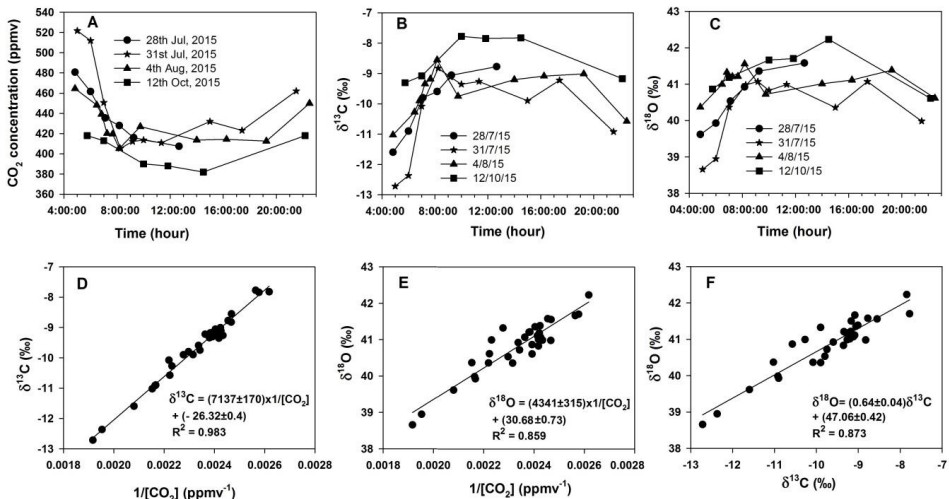


Figure 2. Top panels show the diurnal variation of (A) concentration, (B) $\delta^{13}C$, and (C) $\delta^{18}O$
of $CO_2$ sampled in the greenhouse. Bottom panels are the Keeling plots for (D) $\delta^{13}C$ and (E)
$\delta^{18}O$ and (F) scatter plot of $\delta^{13}C$ and $\delta^{18}O$ to show their covariance.




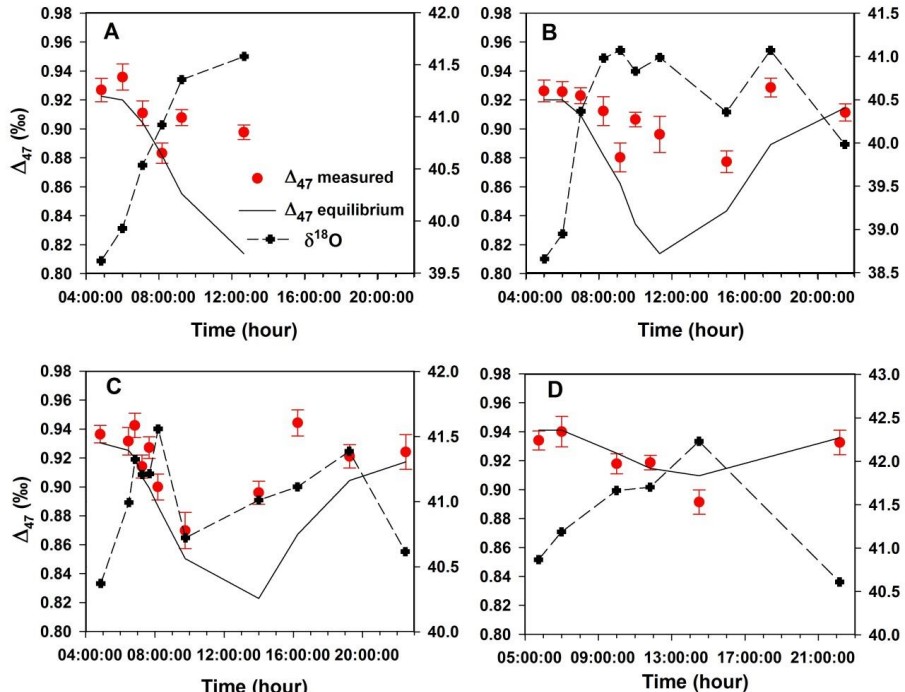


Figure 3. Diurnal variation of the $\Delta_{47}$ and $\delta^{18}O$ values in the greenhouse for samples collected on four days of 2015: (A) 28[th] July, (B) 31[st] July, (C) 4[th] August, and (D) 12[th] October. The first three days (A-C) were bright sunny days and the last one (D) on a cloudy day with covered rooftop (see texts for details). The error bars are 1 standard error associated with the measurements.








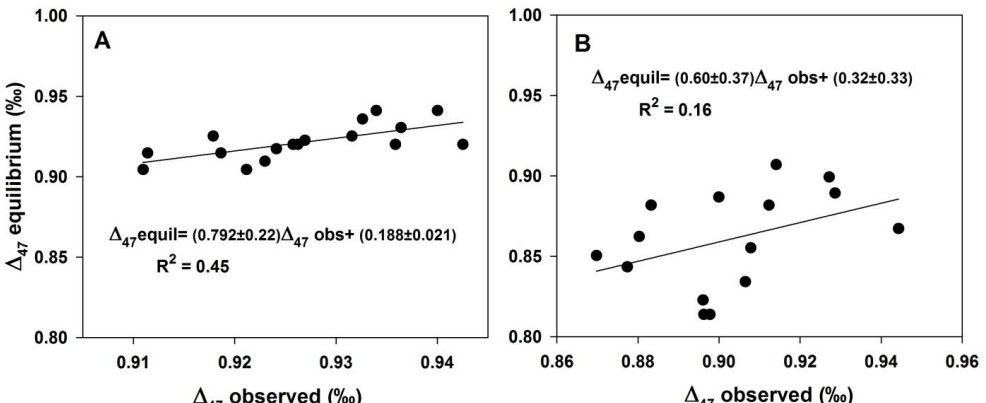


Figure 4. Correlation between the observed and thermodynamic equilibrium $\Delta_{47}$ values for greenhouse $CO_2$ samples collected when (A) photosynthesis is weak and respiration is strong and (B) photosynthesis is strong and respiration is weak.






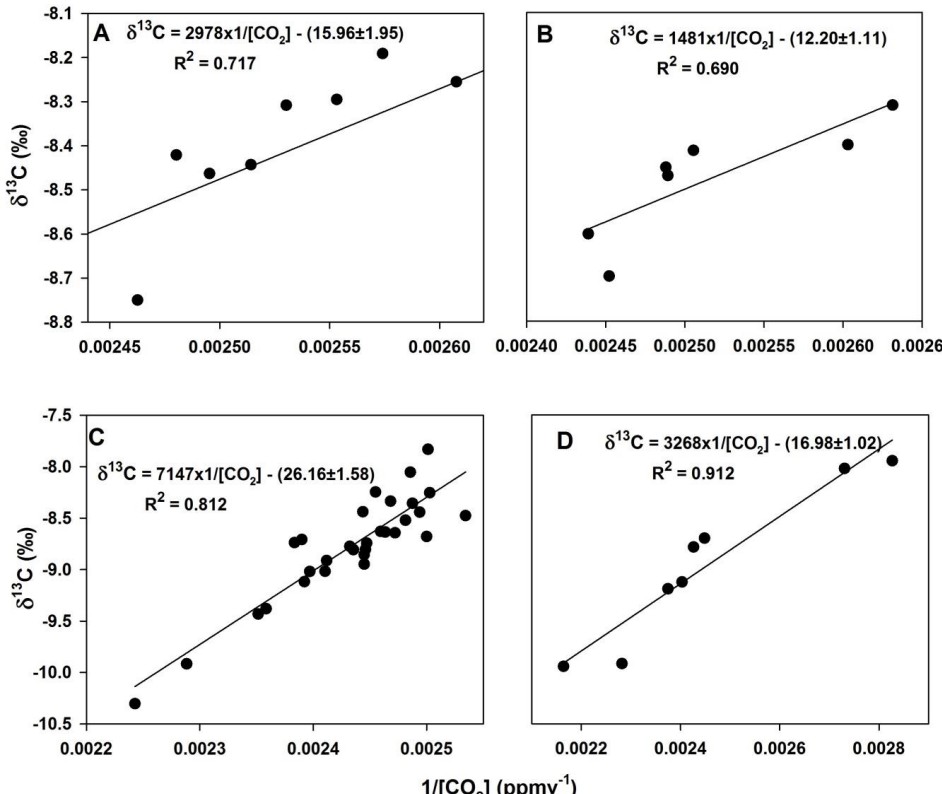

Figure 5. Carbon Keeling plots for atmospheric $CO_2$ collected at (A) South China Sea (B) Keelung and Fuguei Cape, (C) sub-urban station, Academia Sinica Campus, and (D) grassland, National Taiwan University. For more details about the sites, see the texts and Figure 1.





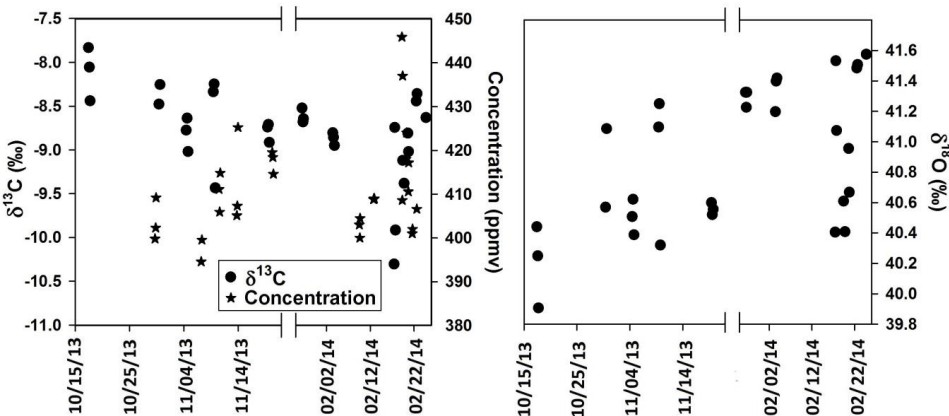


Figure 6. Time series of (A) concentration and stable carbon and (B) stable oxygen isotopes
for $CO_2$ collected at Academia Sinica Campus.



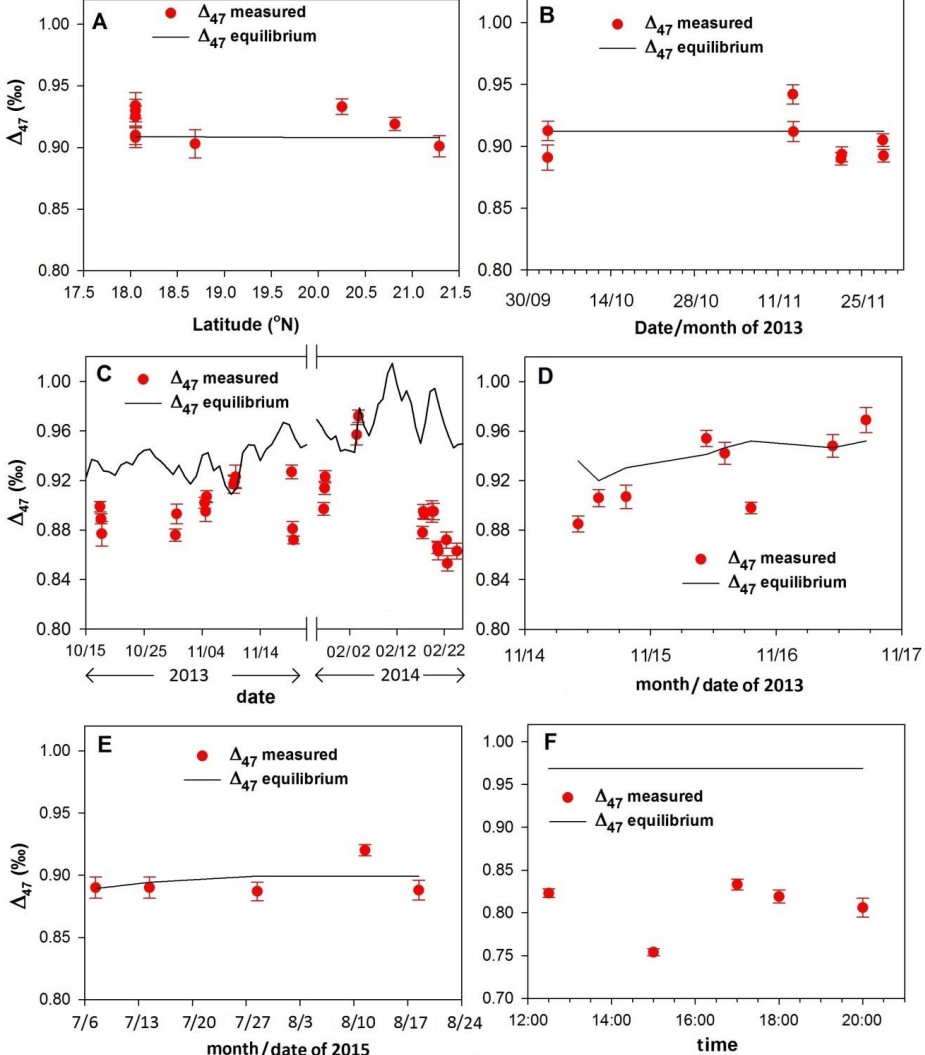

Figure 7. $\Delta_{47}$ values in the near surface atmospheric $CO_2$ from (A) South China Sea, (B) coastal stations (Keelung and Fuguei Cape), (C) sub-urban station (Academia Sinica campus), (D) grassland in the National Taiwan University campus, (E) forest site near the Academia Sinica Campus and (F) urban site (Roosevelt Road). The error bars are the 1 standard errors associated with the measurements. Lines show $\Delta_{47}$ values for the $CO_2$ in thermodynamic equilibrium at ambient temperatures.




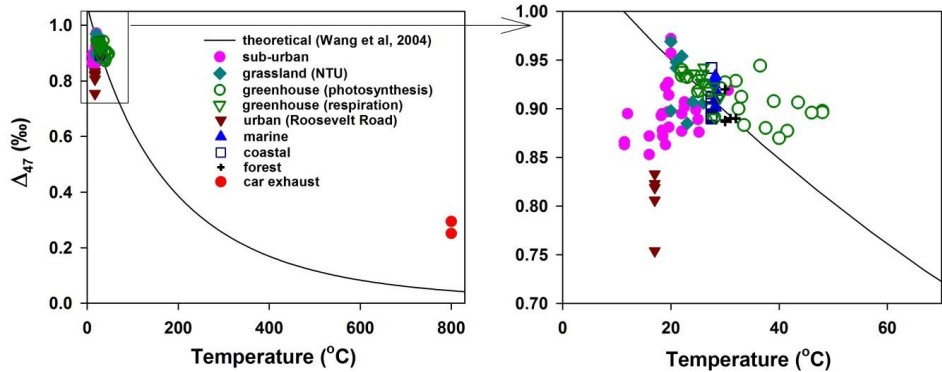



Figure 8. A summary of $\Delta_{47}$ values in near surface air $CO_2$ obtained at different environments
and compared with the thermodynamic equilibrium values. Combustion temperature for car
exhausts is assumed to be 800 $^{o}$C (minimum value). Greenhouse $CO_2$ are divided into two
categories: photosynthesis dominated (green open circle) and respiration dominated (green
open triangle).
















Table 1. Reproducibility and precision of measurements for stable isotopes including $\Delta_{47}$ for IAEA NBS-19.

| Sl. No. | $\delta^{13}C$(‰) (VPDB) | $\delta^{18}O$ (‰) (VPDB) ($CO_2$) | $\delta^{47}$(‰) | Std. Err. | $\Delta_{47}$(‰) | Std. Err. |
|---|---|---|---|---|---|---|
| 1 | 2.02 | -2.21 | 35.22 | 0.01 | 0.382 | 0.010 |
| 2 | 2.02 | -2.11 | 35.54 | 0.02 | 0.394 | 0.012 |
| 3 | 2.02 | -2.19 | 35.28 | 0.01 | 0.416 | 0.010 |
| 4 | 2.01 | -2.28 | 35.15 | 0.01 | 0.408 | 0.011 |
| 5 | 2.00 | -2.27 | 35.24 | 0.02 | 0.388 | 0.016 |
| 6 | 2.00 | -2.16 | 35.27 | 0.02 | 0.370 | 0.013 |
| 7 | 2.02 | -2.27 | 35.21 | 0.01 | 0.398 | 0.009 |
| 8 | 2.02 | -2.20 | 36.48 | 0.01 | 0.363 | 0.008 |
| 9 | 2.01 | -2.20 | 36.56 | 0.02 | 0.392 | 0.006 |
| 10 | 2.01 | -2.15 | 36.46 | 0.01 | 0.399 | 0.012 |
| 11 | 2.01 | -2.20 | 36.57 | 0.01 | 0.393 | 0.010 |
| 12 | 2.02 | -2.21 | 36.32 | 0.01 | 0.387 | 0.009 |
| 13 | 2.01 | -2.18 | 36.43 | 0.01 | 0.368 | 0.014 |
| 14 | 2.01 | -2.16 | 35.81 | 0.01 | 0.379 | 0.010 |
| 15 | 2.00 | -2.18 | 35.76 | 0.01 | 0.387 | 0.006 |
| Average | 2.01 | -2.20 | 35.82 | | 0.388 | |
| Std. Dev. | 0.01 | 0.05 | 0.58 | | 0.014 | |





Table 2. Diurnal variation of $\delta^{13}C$ and $\delta^{18}O$ and clumped isotopes ($\Delta_{47}$) for greenhouse $CO_2$. Temperatures estimated using $\Delta_{47}$ values and actual air temperatures inside the greenhouse at the time of sampling are also presented.

| Date | Time | Conc. (ppmv) | $\delta^{13}C$(‰) (VPDB) | $\delta^{18}O$(‰) (VSMOW) | $\delta^{47}$(‰) | Std. err. | $\Delta_{47}$(‰) (ARF) | Std. err. | Estimated temp. (°C) | Air temp. (°C) |
|---|---|---|---|---|---|---|---|---|---|---|
| 7/28/2015 | 4:50 | 481 | -11.60 | 39.61 | 6.99 | 0.02 | 0.927 | 0.016 | 24 | 25.5 |
| | 6:00 | 462 | -10.90 | 39.92 | 8.16 | 0.02 | 0.936 | 0.018 | 21 | 26 |
| | 7:06 | 435 | -9.80 | 40.54 | 9.71 | 0.02 | 0.911 | 0.017 | 28 | 29 |
| | 8:10 | 428 | -9.60 | 40.92 | 10.38 | 0.02 | 0.883 | 0.014 | 33 | 33.5 |
| | 9:15 | 416 | -9.06 | 41.36 | 11.30 | 0.01 | 0.908 | 0.011 | 24 | 39 |
| | 10:15 | 422 | -9.55 | 40.82 | NA | NA | NA | NA | NA | NA |
| | 12:40 | 407 | -8.77 | 41.58 | 11.75 | 0.01 | 0.898 | 0.010 | 27 | 48 |
| 7/31/2015 | 5:00 | 522 | -12.72 | 38.66 | 5.10 | 0.01 | 0.926 | 0.015 | 24 | 26 |
| | 6:00 | 512 | -12.37 | 38.95 | 5.94 | 0.01 | 0.926 | 0.014 | 25 | 26 |
| | 7:00 | 451 | -10.08 | 40.36 | 9.39 | 0.02 | 0.923 | 0.011 | 25 | 28 |
| | 8:15 | 405 | -8.82 | 40.98 | 11.25 | 0.02 | 0.912 | 0.020 | 28 | 33 |
| | 9:10 | 412 | -9.12 | 41.07 | 11.26 | 0.02 | 0.880 | 0.020 | 34 | 37.5 |
| | 10:00 | 414 | -9.35 | 40.83 | 11.52 | 0.01 | 0.906 | 0.010 | 23 | 43.5 |
| | 11:20 | 411 | -9.26 | 40.99 | 11.12 | 0.02 | 0.896 | 0.025 | 31 | 48 |
| | 15:00 | 432 | -9.90 | 40.36 | 9.55 | 0.02 | 0.877 | 0.015 | 34 | 41.5 |
| | 17:25 | 423 | -9.22 | 41.07 | 12.48 | 0.02 | 0.929 | 0.013 | 25 | 32 |
| | 21:30 | 462 | -10.92 | 39.99 | 7.90 | 0.01 | 0.911 | 0.012 | 28 | 27 |
| 8/4/2015 | 4:50 | 465 | -11.03 | 40.37 | 8.41 | 0.01 | 0.936 | 0.012 | 23 | 24 |
| | 5:50 | 455 | -10.82 | 40.26 | NA | NA | NA | NA | NA | NA |
| | 6:28 | 448 | -10.27 | 41.00 | 10.01 | 0.02 | 0.931 | 0.017 | 24 | 25.5 |
| | 6:50 | 439 | -9.90 | 41.32 | 10.10 | 0.02 | 0.942 | 0.009 | 22 | 26 |




| Time | | | | | | | | | | |
|------|------|--------|-------|-------|------|-------|------|-------|----|------|
| 7:15 | 420 | -9.34 | 41.22 | 11.05 | 0.01 | 0.914 | 0.01 | 0.013 | 28 | 28.5 |
| 7:40 | 419 | -9.18 | 41.22 | 11.05 | 0.01 | 0.927 | 0.01 | 0.011 | 25 | 30 |
| 8:10 | 405 | -8.55 | 41.56 | 12.79 | 0.02 | 0.900 | 0.02 | 0.015 | 31 | 32.5 |
| 9:45 | 427 | -9.75 | 40.73 | 10.81 | 0.02 | 0.870 | 0.02 | 0.023 | 36 | 40 |
| 14:00 | 414 | -9.20 | 41.01 | 11.02 | 0.01 | 0.896 | 0.01 | 0.011 | 31 | 46 |
| 16:15 | 414 | -9.09 | 41.11 | 11.11 | 0.01 | 0.944 | 0.01 | 0.014 | 22 | 36.5 |
| 19:15 | 413 | -9.01 | 41.38 | 13.28 | 0.01 | 0.921 | 0.01 | 0.010 | 26 | 29.2 |
| 22:30 | 450 | -10.58 | 40.61 | 9.34 | 0.02 | 0.924 | 0.02 | 0.022 | 25 | 26.5 |
| 5:45 | 418 | -9.30 | 40.87 | 10.80 | 0.01 | 0.934 | 0.01 | 0.013 | 23 | 22 |
| 7:00 | 413 | -9.08 | 41.18 | 10.95 | 0.02 | 0.940 | 0.02 | 0.021 | 22 | 22 |
| 10:00 | 390 | -7.78 | 41.66 | 13.00 | 0.02 | 0.918 | 0.02 | 0.014 | 26 | 25 |
| 11:50 | 388 | -7.84 | 41.71 | 15.25 | 0.01 | 0.919 | 0.01 | 0.010 | 26 | 27 |
| 14:30 | 382 | -7.82 | 42.24 | 14.27 | 0.02 | 0.891 | 0.02 | 0.017 | 31 | 28 |
| 20:10 | 418 | -9.17 | 40.61 | 10.85 | 0.02 | 0.933 | 0.02 | 0.017 | 23 | 23 |

(10/12/2015)

Table 3. Stable carbon and oxygen isotopic composition and clumped isotopes ($\Delta_{47}$) for car exhaust $CO_2$. Temperatures estimated using $\Delta_{47}$ values and lowest possible combustion temperatures are given.

| Car model | Conc. (ppm) | δ13C(‰) (VPDB) | δ18O(‰) (VSMOW) | δ47(‰) | Std. err. | Δ47(‰) (ARF) | Std. err. | Estimated temp. (°C) | Combustion temp. (°C) |
|-----------|-------------|----------------|-----------------|--------|-----------|--------------|-----------|----------------------|-----------------------|
| Mazda 3000cc TRIBUTE | 39400 | -27.73 | 25.43 | -22.20 | 0.01 | 0.251 | 0.013 | 300 | 800 |
| Mitsubishi 2400cc New Outlander | 39300 | -27.67 | 25.27 | -23.08 | 0.02 | 0.294 | 0.007 | 265 | 800 |
| Average ± 1σ | 39350±50 | -27.70±0.03 | 25.35±0.07 | -22.64±0.44 | | 0.273±0.021 | | 283±18 | |






Table 4. Stable isotopic composition including $\Delta_{47}$ for air $CO_2$ collected over South China Sea and two coastal stations (see Figure 1 for sampling locations). Temperatures estimated using $\Delta_{47}$ values and the sea surface temperatures at the time of samplings are also presented.

**Marine air $CO_2$**

**South China Sea**

| Date time | Conc. (ppm) | $\delta^{13}C$(‰) (VPDB) | $\delta^{18}O$(‰) (VSMOW) | $\delta^{47}$ (‰) | Std. err. | $\Delta_{47}$ (‰) (ARF) | Std. err. | Estimated temp. (°C) | Sea surface temp. (°C) |
|---|---|---|---|---|---|---|---|---|---|
| 10/15/2013 8:15 (A)* | 403 | -8.42 | 40.85 | 28.752 | 0.016 | 0.901 | 0.017 | 30 | 28.3 |
| 10/15/2013 13:15 (B) | 400 | -8.46 | 40.80 | 28.441 | 0.012 | 0.919 | 0.011 | 26 | 28.3 |
| 10/15/2013 18:00 (C) | 406 | -8.75 | 40.54 | 28.133 | 0.013 | 0.933 | 0.013 | 24 | 28.3 |
| 10/16/2013 7:00 (D) | 391 | -8.76 | 40.53 | 27.916 | 0.024 | 0.903 | 0.023 | 29 | 28.2 |
| 10/16/2013 12:05 (E) | 397 | -8.44 | 40.86 | 28.535 | 0.015 | 0.910 | 0.015 | 28 | 28.2 |
| 10/16/2013 14:00 (E) | 391 | -8.30 | 40.96 | 28.922 | 0.021 | 0.934 | 0.021 | 23 | 28.2 |
| 10/16/2013 17:20 (E) | 395 | -8.31 | 41.02 | 28.944 | 0.017 | 0.908 | 0.016 | 29 | 28.1 |
| 10/16/2013 20:20 (E) | 388 | -8.19 | 40.52 | 28.909 | 0.018 | 0.930 | 0.018 | 24 | 28.1 |
| 10/17/2013 8:40 (E) | 383 | -8.26 | 40.41 | 28.194 | 0.018 | 0.925 | 0.018 | 25 | 28.1 |
| Average ± 1σ | 395±7 | -8.43±0.19 | 40.72±0.20 | 28.52±0.36 | | 0.918±0.012 | | 27±2 | 28.2±0.1 |
| **Keelung** | | | | | | | | | |
| 10/03/2013 11:30 | 380 | -8.31 | 40.31 | 28.053 | 0.020 | 0.896 | 0.021 | 31 | 27.5 |
| 10/03/2013 12:30 | 384 | -8.40 | 40.92 | 29.089 | 0.017 | 0.917 | 0.016 | 27 | 27.5 |





| | 401 | -8.45 | 40.62 | 29.645 | 0.015 | 0.946 | 0.016 | 21 | 27.5 |
| 11/13/2013 11:00 | | | | | | | | | |
| 11/21/2013 12:30 | | -8.47 | 40.78 | 29.866 | 0.017 | 0.890 | 0.010 | 32 | 27.5 |
| 11/28/2013 12:00 | 410 | -8.60 | 40.21 | 28.992 | 0.011 | 0.908 | 0.010 | 28 | 27.5 |
| Average ± 1σ | 394±12 | -8.45±0.09 | 40.57±0.26 | 29.12±0.63 | | 0.911±0.020 | | 28±4 | 27.5 |

**Fuguei Cape**

| 11/13/2013 13:30 | 401 | -8.47 | 40.76 | 29.56 | 0.02 | 0.916 | 0.016 | 27 | 27.5 |
| 11/21/2013 15:30 | 399 | -8.41 | 40.89 | 29.37 | 0.01 | 0.880 | 0.012 | 34 | 27.5 |
| 11/28/2013 15:00 | 407 | -8.70 | 41.16 | 30.11 | 0.01 | 0.886 | 0.010 | 33 | 27.5 |
| Average ± 1σ | 402±3 | -8.53±0.12 | 40.94±0.16 | 29.68±0.29 | | 0.894±0.015 | | 31±3 | 27.5 |

*Sampling Stations (see Figure 1 for locations in South China Sea)

Table 5. Stable isotopic composition including clumped isotopes ($\Delta_{47}$) for air $CO_2$ collected in urban and sub-urban stations, grassland, forest and high mountain environments. Temperatures estimated using $\Delta_{47}$ values and air temperatures are also presented.

**Urban $CO_2$: Roosevelt Road, Taipei City**

| Date | Time | Conc. (ppm) | $\delta^{13}C$(‰) (VPDB) | $\delta^{18}O$(‰) (VSMOW) | $\delta^{47}$(‰) | Std. err. | $\Delta_{47}$ (‰) (ARF) | Std. err. | Estimated temp. (°C) | Air temp. (°C) |
|---|---|---|---|---|---|---|---|---|---|---|
| 12/30/ 2015 | 12:30 | 510 | -10.41 | 40.00 | 25.26 | 0.014 | 0.823 | 0.010 | 46 | 20 |
| | 15:00 | 478 | -11.50 | 38.49 | 22.63 | 0.012 | 0.754 | 0.008 | 62 | 19.5 |
| | 17:00 | 461 | -9.69 | 40.70 | 26.74 | 0.017 | 0.833 | 0.013 | 44 | 17 |
| | 18:00 | 594 | -12.30 | 38.14 | 21.56 | 0.014 | 0.819 | 0.015 | 47 | 16 |
| | 20:00 | 457 | -11.34 | 39.24 | 23.61 | 0.022 | 0.806 | 0.022 | 50 | 15 |
| Average±1σ | | 500±50 | -11.05±0.90 | 39.31±0.94 | 23.96±1.84 | | 0.807±0.028 | | 50±6 | 17±2 |




**Sub-urban air CO$_2$**

**Academia Sinica Campus**

| Date time | Conc. (ppm) | $\delta^{13}C$(‰) (VPDB) | $\delta^{18}O$(‰) (VSMOW) | $\delta^{47}$ (‰) | Std. err. | $\Delta_{47}$ (‰) (ARF) | Std. err. | Estimated temp. (°C) | Air temp (°C) |
|---|---|---|---|---|---|---|---|---|---|
| 10/17/2013 10:00 | 400 | -7.83 | 40.44 | 28.47 | 0.015 | 0.899 | 0.008 | 30 | 25 |
| 10/17/2013 14:30 | 402 | -8.05 | 40.25 | 28.07 | 0.017 | 0.889 | 0.008 | 32 | 25 |
| 10/17/2013 17:20 | 409 | -8.44 | 39.90 | 27.26 | 0.019 | 0.877 | 0.020 | 34 | 22 |
| 10/30/2013 10:00 | 395 | -8.48 | 40.57 | 28.47 | 0.012 | 0.876 | 0.010 | 35 | 25.2 |
| 10/30/2013 14:30 | 400 | -8.25 | 41.08 | 29.03 | 0.016 | 0.893 | 0.016 | 31 | 27.4 |
| 11/04/2013 10:30 | 411 | -8.78 | 40.51 | 28.67 | 0.011 | 0.902 | 0.009 | 29 | 22.5 |
| 11/04/2013 14:30 | 406 | -8.64 | 40.62 | 28.97 | 0.017 | 0.895 | 0.016 | 31 | 22 |
| 11/04/2013 18:30 | 415 | -9.02 | 40.38 | 28.33 | 0.013 | 0.907 | 0.009 | 28 | 22.5 |
| 11/09/2013 10:30 | 405 | -8.34 | 41.09 | 29.79 | 0.019 | 0.917 | 0.015 | 27 | 28.5 |
| 11/09/2013 14:00 | 407 | -8.25 | 41.25 | 30.63 | 0.015 | 0.919 | 0.009 | 26 | 30.6 |
| 11/09/2013 18:30 | 425 | -9.43 | 40.32 | 27.49 | 0.020 | 0.923 | 0.019 | 25 | 28 |
| 11/19/2013 10:00 | 419 | -8.74 | 40.60 | 29.27 | 0.012 | 0.927 | 0.011 | 25 | 19.5 |
| 11/19/2013 14:00 | 418 | -8.71 | 40.52 | 29.59 | 0.019 | 0.881 | 0.012 | 33 | 19.6 |
| 11/19/2013 18:00 | 414 | -8.91 | 40.56 | 28.58 | 0.012 | 0.872 | 0.006 | 35 | 18.5 |
| 01/27/2014 10:30 | 403 | -8.52 | 41.32 | 30.13 | 0.008 | 0.897 | 0.010 | 30 | 19.2 |
| 01/27/2014 15:20 | 400 | -8.68 | 41.23 | 30.03 | 0.011 | 0.914 | 0.010 | 27 | 19.6 |
| 01/27/2014 18:00 | 404 | -8.64 | 41.32 | 29.29 | 0.017 | 0.923 | 0.010 | 25 | 18.5 |
| 02/03/2014 11:00 | 408 | -8.80 | 41.20 | 29.67 | 0.015 | 0.957 | 0.017 | 19 | 24.5 |
| 02/03/2014 14:30 | 409 | -8.86 | 41.39 | NA | | NA | | | |
| 02/03/2014 19:30 | 409 | -8.95 | 41.41 | 30.57 | 0.011 | 0.972 | 0.010 | 16 | 19.3 |



| | | | | | | | | | |
|---|---|---|---|---|---|---|---|---|---|
| 02/17/2014 10:30 | 445 | -10.30 | 40.40 | 27.60 | 0.016 | 0.878 | 0.010 | 34 | 22.4 |
| 02/17/2014 14:30 | 408 | -8.74 | 41.53 | 30.58 | 0.014 | 0.895 | 0.011 | 31 | 25 |
| 02/17/2014 18:30 | 437 | -9.92 | 41.07 | 28.49 | 0.012 | 0.893 | 0.008 | 31 | 22 |
| 02/19/2014 10:00 | 418 | -9.12 | 40.61 | 29.12 | 0.020 | 0.895 | 0.018 | 31 | 13.3 |
| 02/19/2014 18:00 | 424 | -9.38 | 40.40 | 28.49 | 0.020 | 0.895 | 0.013 | 31 | 12.4 |
| 02/20/2014 14:30 | 410 | -8.81 | 40.96 | 29.68 | 0.023 | 0.866 | 0.010 | 37 | 12.9 |
| 02/20/2014 18:00 | 417 | -9.02 | 40.66 | 29.59 | 0.018 | 0.863 | 0.014 | 37 | 12.5 |
| 02/22/2014 12:15 | 401 | -8.44 | 41.49 | 30.63 | 0.013 | 0.872 | 0.013 | 35 | 17.5 |
| 02/22/2014 17:00 | 402 | -8.36 | 41.51 | 30.63 | 0.013 | 0.853 | 0.012 | 40 | 17.1 |
| 02/24/2014 17:30 | 406 | -8.63 | 41.57 | 30.70 | 0.014 | 0.863 | 0.013 | 37 | 22 |
| Average ± 1σ | 411±11 | -8.78±0.50 | 40.87±0.46 | 29.23±1.00 | | 0.897±0.027 | | 30±5 | 21±5 |
| **Grassland: NTU Campus** | | | | | | | | | |
| 11/14/2013 10:10 | 353 | -7.95 | 40.96 | 30.18 | 0.02 | 0.885 | 0.013 | 33 | 23 |
| 11/14/2013 14:05 | 366 | -8.02 | 41.31 | 30.79 | 0.01 | 0.906 | 0.014 | 29 | 26 |
| 11/14/2013 19:20 | 462 | -9.94 | 38.33 | 25.64 | 0.02 | 0.907 | 0.019 | 29 | 24 |
| 11/15/2013 10:40 | 416 | -9.12 | 39.42 | 29.51 | 0.01 | 0.954 | 0.013 | 20 | 22 |
| 11/15/2013 14:10 | 421 | -9.19 | 39.36 | 29.78 | 0.02 | 0.942 | 0.018 | 22 | 21 |
| 11/15/2013 19:12 | 438 | -9.92 | 38.28 | 28.08 | 0.04 | 0.989 | 0.009 | 13 | 20 |
| 11/16/2013 10:50 | 412 | -8.78 | 40.03 | 28.54 | 0.02 | 0.948 | 0.018 | 21 | 21 |
| 11/16/2013 17:10 | 408 | -8.70 | 40.26 | 26.06 | 0.02 | 0.969 | 0.021 | 17 | 20 |
| Average ± 1σ | 409±33 | -8.95±0.70 | 39.74±1.00 | 28.57±1.77 | | 0.937±0.030 | | 23±6 | 22±2 |
| **Forest site near Academia Sinica Campus** | | | | | | | | | |
| 07/07/2015 10:30 | 411 | -9.07 | 41.43 | 11.54 | 0.01 | 0.890 | 0.017 | 32 | 32 |
| 07/14/2015 10:30 | 458 | -10.43 | 39.74 | 9.01 | 0.02 | 0.890 | 0.017 | 32 | 31 |
| 07/28/2015 10:40 | 441 | -9.99 | 40.86 | 10.07 | 0.02 | 0.887 | 0.015 | 32 | 30 |
| 08/11/2015 10:40 | 448 | -10.46 | 40.09 | 9.50 | 0.01 | 0.920 | 0.009 | 26 | 30 |

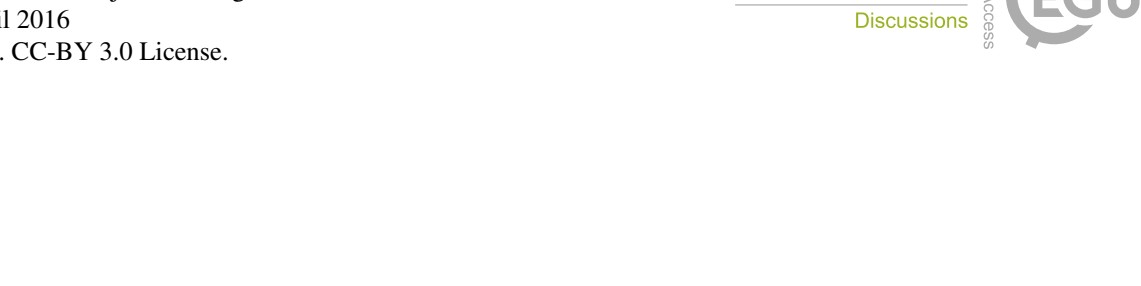

|  |  |  |  |  |  |  |  |  |  |
|---|---|---|---|---|---|---|---|---|---|
| 08/18/2015 10:30 | 433 | -9.99 | 39.80 | 8.99 | 0.02 | 0.888 | 0.016 | 32 | 30 |
| Average ± 1σ | 438±16 | -9.99 ±0.50 | 40.39±0.66 | 9.82±0.94 |  | 0.895±0.012 |  | 31±2 | 31±1 |
| **High mountain: Hehuan** |  |  |  |  |  |  |  |  |  |
| 10/09/2013 13:20 | 364 | -8.21 | 40.89 | 28.79 | 0.02 | 0.895 | 0.016 | 31 | 10 |
| 10/09/2013 17:00 | NA | -8.25 | 40.28 | 28.41 | 0.01 | 0.914 | 0.014 | 27 | 10 |
| Average ± 1σ | 364 | -8.23 ±0.02 | 40.59±0.30 | 28.60±0.19 |  | 0.904±0.009 |  | 30±2 | 10 |

946