# Peer review of "Clumped isotopes in near surface atmospheric CO$_2$ over land, coast and ocean in Taiwan and its vicinity"

_Biogeosciences, 2016_

## Referee Comment (RC1) · Anonymous Referee #3 · 4 May 2016

This manuscript reports new measurements of clumped isotope compositions of atmospheric CO2 collected from different environments and settings. Studies of clumped isotope composition of atmospheric CO2 were among the first applications of clumped isotope methods, but have received less attention in recent years compared to other applications. It's great to see another focused study on this subject. The dataset presented in this study is quite extensive, and mostly confirms the major findings from previous studies. However, the conclusion the authors draw regarding the effect of photosynthesis on the clumped isotope composition of CO2 differs significantly from previous studies, and could potentially open many research opportunities. Overall, this manuscript improves our understanding of the various controls on the clumped isotope

composition of atmospheric CO2, and can help future efforts to better constrain the atmospheric CO2 budgets. I have several specific comments about this manuscript, as detailed below, and would recommend these issues be addressed prior to publication.

Major comments:

1. Separation of N2O from CO2. A GC column was used to separate N2O from CO2 in this study. The authors showed a reasonable separation of the two in Fig. S2, but didn't mention the exact CO2 trapping time in their experiments. It's possible the CO2 yield was compromised in order to achieve the optimal separation of N2O. The authors need to provide more details and discuss how the compromised yield and/or residual N2O might affect their clumped isotope data.

2. Photosynthesis effect. In their greenhouse experiments, the authors observed that the clumped isotope compositions of CO2 were higher than what expected from thermodynamic equilibrium when photosynthesis was active. This finding is very intriguing and differs from what observed in previous studies (e.g. Eiler and Schauble 2004), where the clumped isotope compositions of CO2 residual to photosynthesis were shown to generally decrease.

a. Given the importance of this finding, I think the authors need to provide D48 and D49 data of their measurements to show that the elevated D47 values were not related to any contamination issues. More generally, the authors are encouraged to include all their raw clumped isotope measurement data in the electronic supplementary material of their manuscript, which is becoming a convention in the clumped isotope community.

b. The authors need to expand their discussion about the clumped isotope effects associated with photosynthesis they observed, especially in relation to the findings in Eiler and Schauble (2004), and explore ways to reconcile the findings from the two studies.

c. The authors did a nice job estimating the carbon and oxygen isotope fractionations

associated with photosynthesis in their greenhouse experiments. But their discussion about the clumped isotope effect is mostly qualitative. The authors might want to construct a simple (semi-)quantitative model to simulate the evolution of the concentration and isotopic composition of $CO_2$ in their greenhouse experiments. Such a model might enable them to quantitatively estimate the clumped isotope effects associated with photosynthesis, which would be an important contribution of this study.

Minor comments:

1. Line 440: the authors neglected the daytime respiration when estimating the isotope effects associated with photosynthesis. They need to provide evidence to support this approach.

2. In section 4.1, the authors estimated the rates of respiration, photosynthesis, and $CO_2$-water exchange in their greenhouse experiments, in the unit of molecules cm-2 s-1. But it's not entirely clear how those values were derived. More details are needed.

---

## Referee Comment (RC2) · Anonymous Referee #2 · 10 May 2016

[revised manuscript text omitted]

*Sampling Stations (see Figure 1 for locations in South China Sea)

[revised manuscript text omitted]

---

## Referee Comment (RC3) · Anonymous Referee #1 · 13 May 2016

This study provides excellent dataset for almost all of CO2 isotopologues in the atmosphere. Air samples were collected quite extensively, from open ocean, coasts, mountain, forest, grassland, sub-urban and urban traffic. Moreover, closed terrarium experiment and collecting exhaust from cars were conducted as well. Research plan and obtained results are very nice.

While authors provides very valuable dataset, the individual discussion seems not always nice. My major comments on their discussion are; 1) They apply Keeling plot to most cases for source identification. If the case is simple two-source mixing, Keeling plot must be effective. However, this is generally not applicable for the case that source and sink coexist, except that both are the same isotopic composition (fractionation) and

fluxes. I guess greenhouse experiment and grassland observation may be the cases. When Keeling analysis does work well, then authors seek the reason of inconsistency and develop some discussion. Some of these discussions are not so effective. Authors should pay attention that Keeling plot is not a universal tool. 2) On a related matter of 1), developed discussions about D47 results are mostly concluded to "unknown" enzymatic reaction during photosynthesis. Therefore, any quantitative discussion, such as estimating individual fluxes from/to the urban CO2, is not offered. Another approaches may be possible, I guess.

I think this manuscript is worth-publishing to the journal Biogeosciences after addressing specific comments supplied as a separate file. Specific comments involve these issues, too.

Please also note the supplement to this comment:
http://www.biogeosciences-discuss.net/bg-2016-106/bg-2016-106-RC3-supplement.pdf

———————————————————

[Figure]

**Supplement:**

Specific comments.

L56, 85, 110, 301, 427, 482, 583: Authors used a term "bulk" for d13C and d18O, implying that clumped isotope (D47) is not bulk isotopic composition. To my knowledge, the term "bulk" is often used to distinguish between "weighted-average (bulk) isotope ratio of a material" and "compound-specific isotope ratio of a material", such as d13C for "organic matter" versus "protein, lipids, sugar, etc."; or "weighted-average (bulk) isotope ratio of a compound" versus "position-specific isotope ratio of a compound" such as d13C of long-chain hydrocarbon for "all carbon" or "1,2,3,4,..,n th carbon". In this sense, D47 of CO2 is also "bulk", or D47 is neither part of d13C or d18O but a integration of d13C and d18O to some extent. Thus I think authors should avoid using the term "bulk" for d13C and d18O. Instead, "conventional", "traditional" or without any adjective may be better.

L68: "Evapotranspiration" should be replaced to "transpiration."

L68-69: This sentence is out of context. I guess it may follow the sentence of L58-62.

L76-79: Contextually, this sentence should describe about d18O. But this sentence mentions general characteristics of CO2, not only d18O. Revise or move it to more appropriate place. In addition, I request one or more references to mention that present biogeochemical models remain inconclusive.

L85-87: "..limited because of the challenge.." Somewhat strange. "..limited due to the demand of very high precision.." or "..limited but several challenges have conducted to apply it to the atmospheric study.." might be more suitable.

L92: "..have similar time-scales for the isotope exchange between CO2 and water.."

L91-94: I agree that effect of photosynthesis and respiration on clumped isotope has not been studied well, but I disagree that corresponds to d18O as well. At least, their backgrounds are not equal.

L108-109: One or more references are necessary.

L117-119: This is concluding remark. Move it to conclusion.

L123: Delete "amu"

L130: 2 L; 2 atmospheric pressure

L130-133: I could not understand collection procedures well. Was the flask flushed out prior to sample collection without dehumidifier before collection? I believe such kind of pre-process for flushing should be done with identical condition to sample collection. How long did you take for actual sample collection except for pre-flushing?

L139: What is "systematic analyses"?

L141-142; L146-147: Just to recommend, "..5 m high. It was closed at least one day before each experiment and the ventilation was kept as minimum as possible."

L150-155: Add the height of the canopy.

L155-157: Add each sampling height above sea level.

L169-174: If you used a vacuum line, add which process is in vacuo. If not, I'm sorry.

L194-196: Specify the names of the standard (VPDB, VSMOW, etc., for each).

L212: What is "this limit"?

L217-219: Just to recommend, "Masses 48 and 49 were monitored to confirm isobaric interferences due to contamination of hydrocarbons (Ghosh..).

L221-233: Refer Yoshida et al. (2013) RCM27, 207-215, for the evidence of independence from d47 on D47.

L235, 237: I am not so sure whether this term is really appropriate or not, however "empirical transfer function" is often based on the field observation, such as marine foraminifer community structure versus habitat temperature. Authors obtained a relation experimentally, thus I think "reference frame equation", "laboratory equation" or "local equation" should be more appropriate instead of empirical transfer function.

L237-239: Authors need not to discuss in detail, but should compare their results with former study.

L245: The 1-sigma values of d13C and d18O seem too large whereas that of d47 seems in agreement with previous studies (Table S1). Huntington et al. (2009) described that d13C or d18O uncertainties were roughly an order of magnitude better than d47, because of those higher abundance. Actually, Yoshida et al. (2013) showed these lower uncertainties accordingly. To my knowledge, in any way, if one measures d13C with [44] signal of 12V and integration time of 2.5 hour, the standard deviation may be better than 0.01 permil, not only for single gas but also for several aliquots. Actually, results of CO2 digested from carbonates (Table 1) are similar accordingly. Do you have any idea why uncertainties of cylinder CO2 became so high, or d47 uncertainty became lower relatively?

L250-252: Add references for demonstrating poor consensus.

L254-255: Not only showing deviations from expected temperature, specify the reproducibility of D47 thermometry.

L267-272: Lack of data source of temperature at South China Sea.

L276: Diurnal variation..

L282: Define Keeling plot and describe its purpose before the first use for readers from different fields.

L288: What is expected (potential) contamination of anthropogenic CO2 in the greenhouse?

L296: What does "daytime" correspond? Daytime on 12th October? Or other three days?

L297-299: The criteria of separation between weak/strong for photosynthesis or respiration in Fig. 4 is quite unclear. It seems very arbitrary. Define it clearly, otherwise delete this sentence and Fig. 4.

3.2: Catalytic converter in the exhaust plays a role to convert CO to $CO_2$. Is there any possibility this catalytic reaction may affect d18O value as same as D47, not only by exchanging oxygen with water?

3.3: This section should be divided into each field and reorganize to avoid confusion. For example, marine (including SCS and coastal sites), urban (Roosevelt Road), sub-urban (AS), grassland (NTU) and mountain. I guess authors might confuse a bit. For example, the relations between $CO_2$ ($1/CO_2$) and d18O as well as d13C and d18O for grassland are significant (regressions were done with data from Table 5), unlike its statement found in L346-349.

Incidentally, the order to explain d13C and d18O results is marine, urban, sub-urban, grassland, mountain then forest. On the other hand, that to explain D47 results is marine, sub-urban, grassland, forest, mountain then urban. Easy to confuse.

L314-328, L368-372: These should be reorganized as a separate section "marine CO2" for example.

L330-333, L386-390: These should be reorganized as a separate section "urban CO2" for example.

L333-339, L372-376: These should be reorganized as a separate section "sub-urban CO2" for example.

L339-349, L376-379: These should be reorganized as a separate section "grassland CO2" for example.

L353-357, L379-384: These should be reorganized as a separate section "forest CO2" for example.

L314-328: The analysis based on the Keeling plot and subsequent source identification may be problematic. First, authors did not clarify whether the ocean of the study area/period is source or sink of $CO_2$. Second, data range both of $CO_2$ and d13C are narrow and number of data is limited, thus intercept of regression line must have large uncertainties. Therefore, some sentences from L324 to 328 and associated discussion in Section 4 may not be so meaningful. Moreover, authors should consider marine air interacts with ocean surface layer

(mixed layer), not with deep ocean directly. The inconsistency between opaque Keeling intercept and d13C value from unconnected deep ocean is not surprising at all.

L331: 39.32 instead of 39.319

L332-333: The average d18O value is not different significantly from that of grassland, thus this explanation is partly incorrect.

L344-345: I agree with this conclusion, however not by the result from Keeling plot, but by strong relations of CO2-d18O and d13C-d18O as mentioned above. D47 result may support this, thus I would like to emphasize that all results from same field should be described at once (in same block), should not be separated. However, this kind of concluding remark is supposed to be in the discussion.

L346-349: I totally disagree with this sentence. Authors should verify data again.

L358-367: This block and Fig. 6 may not be necessary.

4: The section and order of description is inconsistent with Results. This prevents readers from moving on smoothly. Consider above mentioned comment and reorganization.

L400-418: These blocks should move to introduction.

L422: "biological" instead of "biogeochemical"

L437-446: The obtained fractionation factor of -15.3, which is significantly different from expected C3-type fractionation, clearly demonstrated that this calculation is not applicable to the photosynthesis-respiration coexisting process. Authors should consider the different approaches. For example, assuming constant respiration rate for whole day (applying night time respiration rate to daytime), then obtaining gross productivity.

L446-454: Describe how consistent with previous studies. Consider same calculation mentioned above.

L455-489: Authors demonstrated that d18O of respired CO2 is out of equilibrium with ambient temperature (water is supposed to have constant value, thus disequilibrium is due to temperature variation). If so, D47 of respired CO2 must be always out of equilibrium as well unless d13C is disequilibrium in a same manner (difficult to postulate due to the different fractionation process).

However, authors mentioned that respired CO2 is in equilibrium with temperature because data in the early morning or night-time show close to equilibrium. This is a contradiction in principle. With keeping this contradiction, authors developed further discussion with respect to catalytic reaction. I cannot say whether the discussion is correct or not, however, I can say authors ignores a significant contradiction in the same block. Temperature change during night-time and cloudy (sun-shaded in addition) daytime were small whereas sunny days had wide range of temperature. Simply considering, larger magnitudes of disequilibrium during

sunny daytime may be attributed this large temperature variation. Alternatively or additionally, authors had better consider that air temperature may be different from body temperature inside leaves. Plants have homeostatic function with respect to temperature, a transpiration. $CO_2$ is respired inside the leaf in partial isotope equilibrium with body temperature, not ambient temperature. I believe authors could develop much more deep and quantitative discussion with data shown in this study, before measuring clumped isotope of $O_2$.

L469: Remove "we believe"

L474: Yeung et al., 2015).

L490-498: This block should move to Summary.

L501-513: As mentioned above, I find it difficult to understand why authors would like to link atmospheric $CO_2$ to respired $CO_2$ in the deep ocean. I think this is unnecessary, and recommend to remove entire this block.

4.3: As mentioned above, authors had better consider the possibility of catalytic reaction between CO and $CO_2$ at the converter.

4.4: Authors gave f, anthropogenic contribution, in the two-source component equation from the difference between observed (urban) and marine $CO_2$. This assumption ignores photosynthetic uptake or influence of other sources completely. Authors should get f by solving simultaneous equations based on the concentration and isotopic composition, conversely, then discuss. This approach may be more purposeful, quantitative and premised (why isotope study is needed).

5 or new 4.6: A trial to estimate individual fluxes of combustion, respiration and photosynthesis for C3 and C4, respectively, from/to the urban (or sub-urban) $CO_2$ is very welcome by using [$CO_2$], d13C, d18O and D47.

Fig. 1: Detail map of collection site in the Taipei city is desirable instead of right panel. Coastal and mountain sites can be involved into the left panel.

Fig. 3C: Although there appears a fair negative relation between d18O and D47 in Figs. 3A, B and D, coordinated rapid drops subsequent increases of these values are found on 4th August (3C) as well as 31st July. Do you have any idea what happened at these periods?

Fig. 4: As mentioned above, the criteria to separate A and B is unclear.

Fig. 5: Data from urban site should be added. Ocean and coastal sites can be merged.

Fig. 6: This figure is unnecessary (see above).

Fig. 7: Reorganize (rearrange) according to the order of results and discussions.

New Fig. 9?: The summarizing diagram for individual fluxes (schematic box diagram) is welcome.

Table 2: Add relative humidity if available.

---

## Author Comment (AC1) · 11 Jul 2016

Dear Editor,

On behalf of all co-authors, I would like to thank you very much for handling the paper and the reviewers for their comments which help us to improve the manuscript in terms of presentation as well as scientific contents. We very much appreciate the effort and amount of time spent by the reviewers to review this manuscript. We agree with most of the concerns of the reviewers and addressed in the revised manuscript. Three major suggestions by the reviewers are as follows:

1. Clumped isotope effect associated with photosynthesis especially in relation to the findings in Eiler and Schauble (2004) and explore the ways to reconcile the findings from the two studies.
2. Application of Keeling plot for source identification when the source and sink coexist. Discussions about D47 results are mostly concluded to "unknown" enzymatic reaction during photosynthesis. Therefore, any quantitative discussion, such as estimating individual fluxes from/to the urban CO2, is not offered.
3. The discussion about the clumped isotope effect is mostly qualitative. The reviewer suggested to construct a simple (semi-) quantitative model to simulate the evolution of the concentration and isotopic composition of CO2 in the greenhouse experiments.

Our response to the first query is as follows: We will elaborately discuss the effect of photosynthesis on the clumped isotope signatures in the residual CO2 and compare our findings with that of the Eiler and Schauble (2004). We are gathering more data at leaf level which will help to understand the effect of photosynthesis on clumped isotopes, and the results will be presented in a future publication. Meanwhile we will link more the ambient CO2 results to the greenhouse data which we have learned a lot more thank to a better controlled environment.

To reply to the second query, we agree with the reviewer that the identification of source using Keeling plot in the cases where both source and sinks co-exists is not valid. However, in most of the cases either source or sink is dominant. e.g., in the case of greenhouse, day time is dominated by photosynthesis and night time by respiration. In the revised manuscript, we will also take the photosynthesis into account and do the appropriate calculation to identify the source.

To reply the third query, we want to assure that more in-depth discussion will be included in the revised manuscript. Some simple modelling works were carried out with the traditional isotopes; we will extend this for clumped isotopes in the revised manuscript.

We have also made substantial modifications following reviewers' comments and suggestions. We already published another manuscript recently where we discussed all details about the data quality, details about the clumped isotope measurements including CO2 purification and cited at appropriate places.

Below, please find our point-by-point response to referee's comments (referee's comments are in italics).

Sincerely yours,

Mao-Chang Liang
Academia Sinica

**Anonymous Referee #1**
*This study provides excellent dataset for almost all of CO2 isotopologues in the atmosphere.*
*Air samples were collected quite extensively, from open ocean, coasts, mountain, forest, grassland, sub-urban and urban traffic. Moreover, closed terrarium experiment and collecting exhaust from cars were conducted as well. Research plan and obtained results are very nice. While authors provides very valuable dataset, the individual discussion seems not always nice. My major comments on their discussion are; 1) They apply Keeling plot to most cases for source identification. If the case is simple two-source mixing, Keeling plot must be effective. However, this is generally not applicable for the case that source and sink coexist, except that both are the same isotopic composition (fractionation) and fluxes. I guess greenhouse experiment and grassland observation may be the cases. When Keeling analysis does work well, then authors seek the reason of inconsistency and develop some discussion. Some of these discussions are not so effective. Authors should pay attention that Keeling plot is not a universal tool. 2) On a related matter of 1), developed discussions about D47 results are mostly concluded to "unknown" enzymatic reaction during photosynthesis. Therefore, any quantitative discussion, such as estimating individual fluxes from/to the urban CO2, is not offered. Another approaches may be possible, I guess.*

We thank the reviewer for appreciating the data. We agree with the points raised by the reviewer and answered the query at the beginning where we summarised the major points. We will provide a detail assessment in the revised manuscript.

*I think this manuscript is worth-publishing to the journal Biogeosciences after addressing specific comments supplied as a separate file. Specific comments involve these issues, too.*
*Please also note the supplement to this comment:*
*http://www.biogeosciences-discuss.net/bg-2016-106/bg-2016-106-RC3-*
*supplement.pdf*

We agree with most of the comments and modified the manuscript accordingly. Point to point reply of the queries of the reviewer are given below.

*L56, 85, 110, 301, 427, 482, 583: Authors used a term "bulk" for d13C and d18O, implying that clumped isotope (D47) is not bulk isotopic composition. To my knowledge, the term "bulk" is often used to distinguish between "weighted-average (bulk) isotope ratio of a material" and "compound-specific isotope ratio of a material", such as d13C for "organic matter" versus "protein, lipids, sugar, etc."; or "weighted-average (bulk) isotope ratio of a compound" versus "position-specific isotope ratio of a compound" such as d13C of long-hain*
*hydrocarbon for "all carbon" or "1,2,3,4,..,n th carbon". In this sense, D47 of CO2 is also "bulk", or D47 is neither part of d13C or d18O but a integration of d13C and d18O to some extent. Thus I think authors should avoid using the term "bulk" for d13C and d18O. Instead,*
*"conventional", "traditional" or without any adjective may be better.*

We agree with the terminology of the reviewer and modified in the revised manuscript.

*L68: "Evapotranspiration" should be replaced to "transpiration."*

Done

*L68-69: This sentence is out of context. I guess it may follow the sentence of L58-62.*

Modified in the revised manuscript

*L76-79: Contextually, this sentence should describe about d18O. But this sentence mentions general characteristics of CO2, not only d18O. Revise or move it to more appropriate place. In addition, I request one or more references to mention that present biogeochemical models remain inconclusive.*

We agree with the reviewer, this is a general statement and removed from the revised manuscript

*L85-87: "..limited because of the challenge.." Somewhat strange. "..limited due to the demand of very high precision.." or "..limited but several challenges have conducted to apply it to the atmospheric study.." might be more suitable.*

The sentence is modified

*L92: "..have similar time-scales for the isotope exchange between CO2 and water.."*

The sentence is modified

*L91-94: I agree that effect of photosynthesis and respiration on clumped isotope has not been studied well, but I disagree that corresponds to d18O as well. At least, their backgrounds are not equal.*

We agree with the reviewer about d18O and modified the sentence accordingly

*L108-109: One or more references are necessary.*

Additional references are provided

*L117-119: This is concluding remark. Move it to conclusion.*

Done

*L123: Delete "amu"*

Done

*L130: 2 L; 2 atmospheric pressure*

Done

*L130-133: I could not understand collection procedures well. Was the flask flushed out prior to sample collection without dehumidifier before collection? I believe such kind of pre-rocess for flushing should be done with identical condition to sample collection. How long did you take for actual sample collection except for pre-flushing?*

Yes, flasks were flushed out prior to sampling for ~10 minutes and flushing was done through the perchlorate (dehumidifier) column. The flasks were equipped with two stopcocks and after flushing the end stopcock was closed and allowed the pressure to build to 2 atm and then isolated by closing the other stopcock. This is described briefly in the revised manuscript. We also refer the details to our previously published papers such as Liang and Mahata (2015).

*L139: What is "systematic analyses"?*

"Systematic" refers to the study performed systematically, i.e., more regular and intensive sampling. To remove possible confusion, the word systematic is removed.

*L141-142; L146-147: Just to recommend, "..5 m high. It was closed at least one day before each experiment and the ventilation was kept as minimum as possible."*

Done

*L150-155: Add the height of the canopy.*

Done

*L155-157: Add each sampling height above sea level.*

Done

*L169-174: If you used a vacuum line, add which process is in vacuo. If not, I'm sorry.*

Yes, we used vacuum line. $CO_2$ was extracted from air using a glass vacuum line connected to a turbo molecular pump by cryogenic technique. The vacuum line as well as the sample flask connection assembly including its head space was pumped to high vacuum before starting the $CO_2$ extraction. The details are mentioned in the revised manuscript. We also refer the details to our previously published papers such as Liang and Mahata (2015).

*L194-196: Specify the names of the standard (VPDB, VSMOW, etc., for each).*

Done

*L212: What is "this limit"?*

The limit here refers to the full scrambling state. In this revised version, we replaced the term by "random distribution."

*L217-219: Just to recommend, "Masses 48 and 49 were monitored to confirm isobaric interferences due to contamination of hydrocarbons (Ghosh..).*

Modified the sentence

*L221-233: Refer Yoshida et al. (2013) RCM27, 207-215, for the evidence of independence from d47 on D47.*

Dependence of d47 on D47 varies from mass spectrometer to mass spectrometer. Therefore, this is not relevant, we will discuss this in the supplementary of the revised manuscript.

*L235, 237: I am not so sure whether this term is really appropriate or not, however "empirical transfer function" is often based on the field observation, such as marine foraminifer community structure versus habitat temperature. Authors obtained a relation experimentally, thus I think "reference frame equation", "laboratory equation" or "local equation" should be more appropriate instead of empirical transfer function.*

Though "empirical transfer function" is used by Dennis et al. (2011), we agree with the reviewer that the "reference frame equation" is more appropriate.

*L237-239: Authors need not to discuss in detail, but should compare their results with former study.*

The reference frame equation varies between mass spectrometer to mass spectrometer, even it differs for a given mass spectrometer at different time. It is known to the community.

*L245: The 1-sigma values of d13C and d18O seem too large whereas that of d47 seems in agreement with previous studies (Table S1). Huntington et al. (2009) described that d13C or d18O uncertainties were roughly an order of magnitude better than d47, because of those higher abundance. Actually, Yoshida et al. (2013) showed these lower uncertainties accordingly. To my knowledge, in any way, if one measures d13C with [44] signal of 12V and integration time of 2.5 hour, the standard deviation may be better than 0.01 permil, not only for single gas but also for several aliquots. Actually, results of CO2 digested from carbonates (Table 1) are similar accordingly. Do you have any idea why uncertainties of cylinder CO2 became so high, or d47 uncertainty became lower relatively?*

For carbonates, it is possible to achieve a std. dev. of 0.01 (Table 1). For air CO2 (compressed cylinder air or atmospheric air), handling/purification worsen the precision. Though efforts have been put (see Liang and Mahata, 2015, for example), the best precision we can get so far for d13C and d18O is ~0.05 per mil. The precision we agree that is not sufficient for CO2 long term monitoring, but is sufficient for the current study. Possible cause is likely that slight fractionations during the extraction cause this variation in d13C and d18O. However, this possible fractionation does not impair the D47 analysis.

*L250-252: Add references for demonstrating poor consensus.*

Dennis et al., (2011), the inter-laboratory comparison shows D47 values NBS-19 from 0.373 to 0.404‰ for three laboratories, reference added in the revised manuscript.

*L254-255: Not only showing deviations from expected temperature, specify the reproducibility of D47 thermometry.*

Done

*L267-272: Lack of data source of temperature at South China Sea.*

Actual measurements during sample collection, mentioned in the revised manuscript.

*L276: Diurnal variation..*

Corrected

*L282: Define Keeling plot and describe its purpose before the first use for readers from different fields.*

A brief description of Keeling plot and purpose is incorporated in the revised manuscript.

*L288: What is expected (potential) contamination of anthropogenic CO2 in the greenhouse?*

The potential contaminants are the ambient air with significant anthropogenic components which was found absent from [CO2] and all the isotope signatures.

*L296: What does "daytime" correspond? Daytime on 12th October? Or other three days?*

It is from morning 9 am to evening 5 pm, statement is modified in the revised manuscript.

*L297-299: The criteria of separation between weak/strong for photosynthesis or respiration in Fig. 4 is quite unclear. It seems very arbitrary. Define it clearly, otherwise delete this sentence and Fig. 4.*

By weak photosynthesis we wanted to mean that the photosynthetic activity was reduced artificially. This was done by covering the greenhouse with a double layered black cloth on a dark cloudy day. This is more clearly explained in the revised manuscript.

*3.2: Catalytic converter in the exhaust plays a role to convert CO to CO2. Is there any possibility this catalytic reaction may affect d18O value as same as D47, not only by exchanging oxygen with water?*

The change in d18O in the exhaust was also observed (Sec. 3.2). We are not aware of any process other than exchange of oxygen isotopes between CO2 and condensed water which can cause the change in the d18O or D47 of the exhaust CO2.

*3.3: This section should be divided into each field and reorganize to avoid confusion. For example, marine (including SCS and coastal sites), urban (Roosevelt Road), sub-urban (AS), grassland (NTU) and mountain. I guess authors might confuse a bit. For example, the relations between CO2 (1/CO2) and d18O as well as d13C and d18O for grassland are significant (regressions were done with data from Table 5), unlike its statement found in*
*L346-349. Incidentally, the order to explain d13C and d18O results is marine, urban, sub-urban, grassland, mountain then forest. On the other hand, that to explain D47 results is marine, sub-urban, grassland, forest, mountain then urban. Easy to confuse.*

We reorganized the section and presentation is consistent in the revised manuscript.

*L314-328, L368-372: These should be reorganized as a separate section "marine CO2" for example.*

Done

*L330-333, L386-390: These should be reorganized as a separate section "urban CO2" for example.*

Done

*L333-339, L372-376: These should be reorganized as a separate section "sub-urban CO2" for example.*

Done

*L339-349, L376-379: These should be reorganized as a separate section "grassland CO2" for example.*

Done

*L353-357, L379-384: These should be reorganized as a separate section "forest CO2" for example.*

Done

*L314-328: The analysis based on the Keeling plot and subsequent source identification may be problematic. First, authors did not clarify whether the ocean of the study area/period is source or sink of CO2. Second, data range both of CO2 and d13C are narrow and number of data is limited, thus intercept of regression line must have large uncertainties. Therefore, some sentences from L324 to 328 and associated discussion in Section 4 may not be so meaningful. Moreover, authors should consider marine air interacts with ocean surface layer*

*(mixed layer), not with deep ocean directly. The inconsistency between opaque Keeling intercept and d13C value from unconnected deep ocean is not surprising at all.*

We agree with the reviewer about the application of Keeling plot with a few data points covering a small range. The region is a net source of CO2 in the atmosphere, discussed in the revised manuscript. We put less emphasis on the Keeling plots over the ocean in the revised manuscript.

*L331: 39.32 instead of 39.319*

Done

*L332-333: The average d18O value is not different significantly from that of grassland, thus this explanation is partly incorrect.*

The mean values are different though the uncertainty associated with the values is large. d13C values are significantly different, but it is difficult to conclude based on d18O as mentioned in the later part of the section. The statements are modified in the revised manuscript.

*L344-345: I agree with this conclusion, however not by the result from Keeling plot, but by strong relations of CO2-d18O and d13C-d18O as mentioned above. D47 result may support this, thus I would like to emphasize that all results from same field should be described at once (in same block), should not be separated. However, this kind of concluding remark is supposed to be in the discussion.*

We agree that this should be discussed as a block in the discussion, modified in the revised manuscript.

*L346-349: I totally disagree with this sentence. Authors should verify data again.*

Away from, for example, significant anthropogenic sources, due to presence of a variety of water sources (leaf water, soil water, etc), correlation between 1/[CO2] and d18O is always not observable.

*L358-367: This block and Fig. 6 may not be necessary.*

This paragraph along with Fig. 6 has been removed from the revised manuscript.

*4: The section and order of description is inconsistent with Results. This prevents readers from moving on smoothly. Consider above mentioned comment and reorganization.*

We thank the reviewer for the suggestion. This section is totally reorganized in the revised manuscript.

*L400-418: These blocks should move to introduction.*

Removed from the revised manuscript

*L422: "biological" instead of "biogeochemical"*

Done

*L437-446: The obtained fractionation factor of -15.3, which is significantly different from expected C3-type fractionation, clearly demonstrated that this calculation is not applicable to the photosynthesis-respiration coexisting process. Authors should consider the different approaches. For example, assuming constant respiration rate for whole day (applying night time respiration rate to daytime), then obtaining gross productivity.*

We agree with the reviewer that the calculation should include respiration also. We modified our calculation assuming a constant respiration and presented the estimate in the revised manuscript.

*L446-454: Describe how consistent with previous studies. Consider same calculation mentioned above.*

Calculation is modified as per the suggestion and the calculated d18O discrimination has been compared with previous studies in the revised manuscript.

*L455-489: Authors demonstrated that d18O of respired CO2 is out of equilibrium with ambient temperature (water is supposed to have constant value, thus disequilibrium is due to temperature variation). If so, D47 of respired CO2 must be always out of equilibrium as well unless d13C is disequilibrium in a same manner (difficult to postulate due to the different fractionation process). However, authors mentioned that respired CO2 is in equilibrium with temperature because data in the early morning or night-time show close to equilibrium. This is a contradiction in principle. With keeping this contradiction, authors developed further discussion with respect to catalytic reaction. I cannot say whether the discussion is correct or not, however, I can say authors ignores a significant contradiction in the same block. Temperature change during night-time and cloudy (sun-shaded in addition) daytime were small whereas sunny days had wide range of temperature. Simply considering, larger magnitudes of disequilibrium during sunny daytime may be attributed this large temperature variation. Alternatively or additionally, authors had better consider that air temperature may be different from body temperature inside leaves. Plants have homeostatic function with respect to temperature, a transpiration. CO2 is respired inside the leaf in partial isotope equilibrium with body temperature, not ambient temperature. I believe authors could develop much more deep and quantitative discussion with data shown in this study, before measuring clumped isotope of O2.*

We did not say that d18O of respired CO2 is out of equilibrium. We only showed that the respired CO2 is in thermodynamic equilibrium with the leaf and soil water using the obtained D47 values. We agree with the reviewer that the plant body temperature could be different from the air temperature but with progress of the day we expect change in the D47 values. As stated in the later part of this section, this needs to be tested at leaf level which we are planning and hopefully, the results will help to understand/model the effect of photosynthesis on the D47 values.

*L469: Remove "we believe"*

Done

*L474: Yeung et al., 2015).*

Done

*L490-498: This block should move to Summary.*

Done

*L501-513: As mentioned above, I find it difficult to understand why authors would like to link atmospheric CO2 to respired CO2 in the deep ocean. I think this is unnecessary, and recommend to remove entire this block.*

We agree with the reviewer to remove this paragraph from the revised manuscript

4.3: As mentioned above, authors had better consider the possibility of catalytic reaction between CO and CO2 at the converter.

Yes, reaction between CO and CO2 inside the catalytic converter at the temperature of the converter could also lead to the change in the D47 values, though this would not change in the d18O values as the source of O2 in both CO and CO2 is the atmospheric O2. This is discussed in the revised manuscript.

4.4: Authors gave f, anthropogenic contribution, in the two-source component equation from the difference between observed (urban) and marine CO2. This assumption ignores photosynthetic uptake or influence of other sources completely. Authors should get f by solving simultaneous equations based on the concentration and isotopic composition, conversely, then discuss. This approach may be more purposeful, quantitative and premised (why isotope study is needed).

We agree with the reviewer that a more quantitative estimate for CO2 cycling fluxes between reservoirs is possible. However we note that for example, atmospheric transport, that we mentioned at the end of the section, can easily interfere the calculation (box model interpretation, for example). This is the main reason that we give a more quantitative assessment for the greenhouse data, but not ambient CO2 data.

5 or new 4.6: A trial to estimate individual fluxes of combustion, respiration and photosynthesis for C3 and C4, respectively, from/to the urban (or sub-urban) CO2 is very welcome by using [CO2], d13C, d18O and D47.

Please see the previous response. We agree that the multiple CO2 isotopologues can help to constrain the CO2 fluxes of combustion, respiration and photosynthesis for C3 and C4, etc. However, incomplete knowledge on meteorological influence and lack of systematic dataset around the region prevent us from full assessment. From the available data presented, we showed that D47 behaves differently from [CO2], d13C, and d18O. To minimize regional and/or global interference (due to atmospheric transport, for example), we use greenhouse as a testbed for assessing the associated biological CO2 fluxes. For combustion, there are other tracers more useful than the presented CO2 isotopologues, such as VOCs and 14C.

Fig. 1: Detail map of collection site in the Taipei city is desirable instead of right panel. Coastal and mountain sites can be involved into the left panel.

Done

Fig. 3C: Although there appears a fair negative relation between d18O and D47 in Figs. 3A, B and D, coordinated rapid drops subsequent increases of these values are found on 4th August (3C) as well as 31st July. Do you have any idea what happened at these periods?

Actually the correlation is significant only in Figure 3D. The reason for the rapid decrease in the D47 values in the early in response to photosynthesis is not very clear. We are doing more study at leaf level to identify the possible cause.

Fig. 4: As mentioned above, the criteria to separate A and B is unclear.

Here we wanted to show that D47 values are similar to that expected thermodynamically when respiration is strong and photosynthesis is weak but not the other way round. This is elaborated in the revised manuscript

Fig. 5: Data from urban site should be added. Ocean and coastal sites can be merged.

Done

Fig. 6: This figure is unnecessary (see above).

Removed from the revised manuscript

*Fig. 7: Reorganize (rearrange) according to the order of results and discussions.*

Done

*New Fig. 9?: The summarizing diagram for individual fluxes (schematic box diagram) is welcome.*

We agree that a summarizing diagram of individual flux will enhance the presentation. However, with the present data it will be too early to assign D47 values to individual fluxes. We will keep this suggestion in mind and try in future with more data.

*Table 2: Add relative humidity if available.*

The data from nearest weather stations are added.

**Anonymous Referee #2**
*The manuscript "Clumped isotopes in near surface atmospheric CO2 over land, coast and ocean in Taiwan and its vicinity" provided a valuable dataset of clumped isotopes in atmospheric CO2 and the authors did a good job. For the comments please see the attached file.*

We thank the reviewer for appreciating our effort. All the reviewers queries from the pdf and modifications/changes made in the revise manuscript are listed below. Also the other minor suggestions such as changing present/past tenses in the sentences, deleting/adding texts in the manuscript will be made in the revised manuscript.

*Line 28: The sentences should be in past tense.*

Done

*Line 32: Not clear which processes. mention them i.e. photosynthesis, fossil fuel combustion ...*

The different processes are photosynthesis, respiration, local anthropogenic emissions, modified in the revised manuscript.

*Line 33: Split the sentence*

The sentence is modified

*Line 34: Restructure the sentence: for example, the contribution of various sources of CO2 on D47 ...*

The sentence is restructured
*Line 41: Split the sentence*

Done

*Line 61: Split the sentence because it is hard to follow what you mean. Maybe: ... ocean and landbiosphere. The photosyn... 13C in plants is higher than ... .*

The sentence is divided into two for making it simple and easily understandable

*Line 63: It is not clear what you mean. You should explain how photosynthesis and respiration may change 18O of CO2 in vicinity of plants, if it is what you wanted to say. Is there any discrimination against 18O during assimilation of CO2 for photosynthesis which may lead to enrichment or depletion in CO2 besides the leaves? In the next sentence your*

*explanation just shows enrichement because of evapotranspiration but what is the effect of photosynthesis? Would be this isotopic discrimination due to evapotranspiration against 18O still present if the plant was not under water stress at all?*

The statements are modified as follows:
$\delta^{18}O$ is used for partitioning net $CO_2$ terrestrial fluxes between soil respiration and exchange with the plant leaves, the exchange is enhanced by the presence of carbonic anhydrase in plants and soils (Francey and Tans, 1987; Farquhar and Lioyd, 1993; Yakir and Wang, 1996; Ciais et al., 1997; Peylin et al., 1999; Murayama et al., 2010; Welp et al., 2011). This is because $\delta^{18}O$ of $CO_2$ fluxes originated from soil respiration are different from that exchanged with the leaf water. $\delta^{18}O$ in soil water reflect the $\delta^{18}O$ value of the local meteoric water while leaf water is relatively enriched due to transpiration.

*Line 69: need reference*

Appropriate references are included

*Line 71: This sentence should be in line 62 before 18O is used for partitioning ... .*

Done

*Line 79: You mean reservoirs with different 18O?*

Yes, the statement is modified in the revised manuscript

*Line 85: Split the sentence. You mixed many things together.*

Done

*Line 86-96: Very well! This makes your study unique and valuable.*

We thank reviewer for appreciating the work

*Line 271: Materials and Methods is good.*

Thank you

*Line 277: The lowest CO2 concentration, [CO2] and the highest ...*

Corrected

*Line 296: equilibrium with what? split the sentence.*

Thermodynamic equilibrium with the leaf and soil water, sentence modified and split.

*Line 300: my suggestion: The correlation between D47 and CO2... was observed only when the photosynthesis was weak.*

Suggestion implemented

*Line 302: very good finding.*

Thank you

*Line 399: This paragraph can be deleted. It is not discussing any of the observation and measurements.*

This paragraph has been removed from the revised manuscript

*Line 404: The sentences after "however" are not kind of discussion. I did not get why they should be mentioned here.*

These three paragraphs have been removed from the revised manuscript.

*Line 418: I think the whole these 3 paragraphs should be deleted. It is not clear what you wanted to say. Even if it was like an introduction for the discussion (which is not really necessary) you should follow to emphasis on the main issues respctively to what you will mention later for example effect of photosynthesis on D47, antropogenic effects in urban regions, ... .*

These three paragraphs have been removed from the revised manuscript.

*Line 477: Split the sentence*

Done

*Line 505: put the reference here and split the sentences.*

Done

*Line 507: It is better to mention the intercept value here*

The intercept value is mentioned now

*Line 511: write the value*

Done

Line 521: refer to the fig. or value here

Done

*Line 529: So D47 values in CO2 over oceans at nights should show no deviation from thermodynamic equilibrium. Is that true? How would be this effect in coastal areas where because of shallow water aquous plants may live as well?*

Yes, there should not be any deviation in the D47 in night also. The effect of photosynthesis on clumped isotopes is observable when photosynthesis is very strong e.g., in a confined greenhouse. Probably effect is present everywhere but not detectable with the measurement precision. Therefore, in the coastal areas we expect similar D47 values as observed over the open ocean unless there is a significant CO2 is contributed from the other sources such vehicle and industrial emissions.

*Line 559: It seems logical but how? Do you have an estimation of isotopic composition of condensed water? How CO2 isotopic composition can change? I mean CO2 will dissolve in water but how its isotopic composition can change?*

Unfortunately we do not have any measurement of the d18O value of the condensed water but it is expected to be similar to the atmospheric O2 plus the fractionation associated with the condensation (atmospheric O2 is used for combustion). CO2 readily exchanges oxygen isotopes when comes in contact with water, here probably a partial exchange takes place causing the deviation from the expected d18O and D47 values.

*Line 562: Split the sentence! It is hard to follow you.*

The sentence is split

*Line 566: mention the temperature*

Temperature of the catalytic converter as well as the possible mechanism has been explained in the revised manuscript.

*Line 583: reference needed*

Done

*Line 589: split the sentence*

Done

*Line 604: Can it be also less anthropogenic contribution?*

It can also be due to underestimation of the anthropogenic $CO_2$ at the sampling spot. The regional background [$CO_2$] here could be lower than that assumed and the actual anthropogenic fraction of $CO_2$ could be higher than that assumed here. Discussed in the revised manuscript

*Line 620: split the sentence.*

Done

*Line 625: How could be anthropogenic effects in a dense and isolated forest area?*

It is very unlikely to have anthropogenic CO2 in an isolated place, but we did not neglect a priory. Later using D47 we showed that there is no anthropogenic CO2 in the forest site.

*Line 628: I think still you cannot exclude the photosynthesis effect. Furthermore, the humidity in such condition should be high which may modify the effect of respiration and may affect the temperature. You need to discuss about these issues.*

We agree that we cannot exclude photosynthetic effect. The issue is discussed more elaborately in the revised manuscript.

*Line 648: You need to discuss here about the photosynthetic effect on D47 values, especially that you took the samples just above the grasses.*

We agree with this suggestion and discussed more on this in the revised manuscript.

*Line 657: The fig. 8 is really a good job.*

Thank you

*Line 671: explain it here also how and through which mechanism.*

The explanation is also given here

*Line 678: You did not discuss about this before. Delet this or explain the relation in the text. Why not "atmospheric CO2 budget" instead?*

This is removed from the revised manuscript and instead "atmospheric CO2 budget" is used

**Anonymous Referee #3**
*This manuscript reports new measurements of clumped isotope compositions of atmospheric CO2 collected from different environments and settings. Studies of clumpedisotope composition of atmospheric CO2 were among the first applications of clumped isotope methods, but have received less attention in recent years*

*compared to other applications. It's great to see another focused study on this subject. The dataset presented in this study is quite extensive, and mostly confirms the major findings from previous studies. However, the conclusion the authors draw regarding the effect of photosynthesis on the clumped isotope composition of CO2 differs significantly from previous studies, and could potentially open many research opportunities. Overall, this manuscript improves our understanding of the various controls on the clumped isotope composition of atmospheric CO2, and can help future efforts to better constrain the atmospheric CO2 budgets. I have several specific comments about this manuscript, as detailed below, and would recommend these issues be addressed prior to publication.*

We thank the reviewer for appreciating the work. Effect of photosynthesis on the clumped isotopes of atmospheric CO2 is an interesting finding and will be discussed more elaborately in the revised manuscript. We agree with all the concerns of the reviewer and addressed them in the revised manuscript.

Major comments:
*1. Separation of N2O from CO2. A GC column was used to separate N2O from CO2 in this study. The authors showed a reasonable separation of the two in Fig. S2, but didn't mention the exact CO2 trapping time in their experiments. It's possible the CO2 yield was compromised in order to achieve the optimal separation of N2O. The authors need to provide more details and discuss how the compromised yield and/or residual N2O might affect their clumped isotope data.*

We collect CO2 during 14 – 23 min. Yes CO2 yield was slightly compromised in order to achieve the optimal separation of N2O. The recovery of CO2 was always greater than 95 %. The residual N2O is checked comparing the sample and reference 48 and 49 signals. If value of D48 is large (>5) we did not consider the measurement, either we clean the sample again or just through. When the signal voltage for 49 between sample and reference is more than 0.3 mV we also neglected those numbers. We have discussed this in a recently published article (Laskar et al. Terra Nova, 2016) and cited at appropriate place.

*2. Photosynthesis effect. In their greenhouse experiments, the authors observed that the clumped isotope compositions of CO2 were higher than what expected from thermodynamic equilibrium when photosynthesis was active. This finding is very intriguing and differs from what observed in previous studies (e.g. Eiler and Schauble 2004), where the clumped isotope compositions of CO2 residual to photosynthesis were shown to generally decrease.*

Please see a previous response where the issue is discussed.

*a. Given the importance of this finding, I think the authors need to provide D48 and D49 data of their measurements to show that the elevated D47 values were not related to any contamination issues. More generally, the authors are encouraged to include all their raw clumped isotope measurement data in the electronic supplementary material of their manuscript, which is becoming a convention in the clumped isotope community.*

We will include the raw data including D48 and D49 in the revised manuscript.

*b. The authors need to expand their discussion about the clumped isotope effects associated with photosynthesis they observed, especially in relation to the findings in Eiler and Schauble (2004), and explore ways to reconcile the findings from the two studies.*

Please see a previous response where the issue is discussed.

*c. The authors did a nice job estimating the carbon and oxygen isotope fractionations associated with photosynthesis in their greenhouse experiments. But their discussion about the clumped isotope effect is mostly qualitative. The authors might want to construct a simple (semi-)quantitative model to simulate the evolution of the concentration and isotopic composition of CO2 in their greenhouse experiments. Such a*

*model might enable them to quantitatively estimate the clumped isotope effects associated with photosynthesis, which would be an important contribution of this study.*

Please see the previous response where the issue is discussed.

Minor comments:
*1. Line 440: the authors neglected the daytime respiration when estimating the isotope effects associated with photosynthesis. They need to provide evidence to support this approach.*

We agree that this is a crude assumptions, we will redo the calculations incorporating the day time respiration in the revised manuscript.

*2. In section 4.1, the authors estimated the rates of respiration, photosynthesis, and CO2-water exchange in their greenhouse experiments, in the unit of molecules cm-2 s-1. But it's not entirely clear how those values were derived. More details are needed.*

We will provide more details about the calculations in the revised manuscript.

---

## Author Response (AR2)

Dear Editor,

I would like to thank you very much for the comments which helped us to improve the manuscript. We very much appreciate the effort and amount of time spent by the reviewers to review this manuscript. We agree with most of the concerns of the reviewers and addressed in the revised manuscript. Content wise the revised manuscript is not very different except some changes in the calculations and extending the simple model works to clumped isotope data inside the greenhouse (in Section 4.1) as suggested by the reviewers. The major changes are the presentation, English improvement and rearrangement and restructuring the manuscript as per the reviewers' suggestions.

Three major suggestions by the reviewers are as follows:

1. Clumped isotope effect associated with photosynthesis especially in relation to the findings in Eiler and Schauble (2004) and explore the ways to reconcile the findings from the two studies.
2. Application of Keeling plot for source identification when the source and sink coexist. Discussions about D47 results are mostly concluded to "unknown" enzymatic reaction during photosynthesis. Therefore, any quantitative discussion, such as estimating individual fluxes from/to the urban CO2, is not offered.
3. The discussion about the clumped isotope effect is mostly qualitative. The reviewer suggested to construct a simple (semi-) quantitative model to simulate the evolution of the concentration and isotopic composition of CO2 in the greenhouse experiments.

Our response to the first query is as follows: We elaborately discussed the effect of photosynthesis on the clumped isotope signatures in the residual CO2 and compared our findings with that of the Eiler and Schauble (2004) (see Sec 4.1). We are gathering more data at leaf level which will help to understand the effect of photosynthesis on clumped isotopes and the results will be presented in future publication. Meanwhile we linked more the ambient CO2 results to the greenhouse data which we have learned a lot more thank to a better controlled environment.

To reply to the second query, we agree with the reviewer that the identification of source using Keeling plot in the cases where both source and sinks co-exists is not valid. However, in most of the cases either source or sink is dominant. e.g., in the case of greenhouse, day time is dominated by photosynthesis and night time by respiration. In the revised manuscript, we considered all these aspects and did the appropriate calculation (Sec Sec 4.1).

To reply the third query, we want tell that we have carried out more in-depth discussion in the revised manuscript. Some simple modelling works were carried out with the traditional isotopes; we extended it to the clumped isotope data obtained inside the greenhouse (Sec 4.1 and Figure 8 in the revised manuscript). Definitely this is a new direction and has lot of scopes to carry research in more controlled environment.

I would like to mention that we have recently published a manuscript on clumped isotopes (Laskar et al., 2016, Terra Nova) where we discussed all details about the data quality and clumped isotope measurements including CO2 purification. As a result we have removed some of the contents in the method section and Supplement of the revised manuscript. The published paper is cited at appropriate places.

Below, please find our point-by-point response to referee's comments (referee's comments are in italics).

Sincerely yours,

Mao-Chang Liang
Academia Sinica

**Anonymous Referee #1**
*This study provides excellent dataset for almost all of CO2 isotopologues in the atmosphere. Air samples were collected quite extensively, from open ocean, coasts, mountain, forest, grassland, sub-urban and urban traffic. Moreover, closed terrarium experiment and collecting exhaust from cars were conducted as well. Research plan and obtained results are very nice. While authors provides very valuable dataset, the individual discussion seems not always nice. My major comments on their discussion are; 1) They apply Keeling plot to most cases for source identification. If the case is simple two-source mixing, Keeling plot must be effective. However, this is generally not applicable for the case that source and sink coexist, except that both are the same isotopic composition (fractionation) and fluxes. I guess greenhouse experiment and grassland observation may be the cases. When Keeling analysis does work well, then authors seek the reason of inconsistency and develop some discussion. Some of these discussions are not so effective. Authors should pay attention that Keeling plot is not a universal tool. 2) On a related matter of 1), developed discussions about D47 results are mostly concluded to "unknown" enzymatic reaction during photosynthesis. Therefore, any quantitative discussion, such as estimating individual fluxes from/to the urban CO2, is not offered. Another approaches may be possible, I guess.*

We thank the reviewer for appreciating the data. We agree with the points raised by the reviewer and answered the query at the beginning where we summarised the major points. We provided a detail assessment in the revised manuscript. However, it is too early to estimate individual fluxes using these limited data. Definitely, this will be our next plan with more detailed study including leaf level data.

*I think this manuscript is worth-publishing to the journal Biogeosciences after addressing specific comments supplied as a separate file. Specific comments involve these issues, too. Please also note the supplement to this comment: http://www.biogeosciences-discuss.net/bg-2016-106/bg-2016-106-RC3-supplement.pdf*

We agree with most of the comments and modified the manuscript accordingly. Point to point reply of the queries of the reviewer are given below.

*L56, 85, 110, 301, 427, 482, 583: Authors used a term "bulk" for d13C and d18O, implying that clumped isotope (D47) is not bulk isotopic composition. To my knowledge, the term "bulk" is often used to distinguish between "weighted-average (bulk) isotope ratio of a material" and "compound-specific isotope ratio of a material", such as d13C for "organic matter" versus "protein, lipids, sugar, etc."; or "weighted-average (bulk) isotope ratio of a compound" versus "position-specific isotope ratio of a compound" such as d13C of long-hain hydrocarbon for "all carbon" or "1,2,3,4,..,n th carbon". In this sense, D47 of CO2 is also "bulk", or D47 is neither part of d13C or d18O but a integration of d13C and d18O to some extent. Thus I think authors should avoid using the term "bulk" for d13C and d18O. Instead, "conventional", "traditional" or without any adjective may be better.*

We agree with the terminology of the reviewer and used "conventional" isotopes in the revised manuscript.

*L68: "Evapotranspiration" should be replaced to "transpiration."*
Done (L 74)

*L68-69: This sentence is out of context. I guess it may follow the sentence of L58-62.*

Modified in the revised manuscript (L 63)

*L76-79: Contextually, this sentence should describe about d18O. But this sentence mentions general characteristics of CO2, not only d18O. Revise or move it to more appropriate place. In addition, I request one or more references to mention that present biogeochemical models remain inconclusive.*

We agree with the reviewer, this is a general statement removed from the revised manuscript.

*L85-87: "..limited because of the challenge.." Somewhat strange. "..limited due to the demand of very high precision.." or "..limited but several challenges have conducted to apply it to the atmospheric study.." might be more suitable.*

The sentence is modified (L 84)

*L92: "..have similar time-scales for the isotope exchange between CO2 and water.."*

The sentence is modified (L 90)

*L91-94: I agree that effect of photosynthesis and respiration on clumped isotope has not been studied well, but I disagree that corresponds to d18O as well. At least, their backgrounds are not equal.*

We agree with the reviewer about d18O and modified the sentence accordingly (L 91-94)

*L108-109: One or more references are necessary.*

Additional references are provided (L 107)

*L117-119: This is concluding remark. Move it to conclusion.*

These two sentences are removed from here

*L123: Delete "amu"*

Done (L 119)

*L130: 2 L; 2 atmospheric pressure*

Done (L 126)

*L130-133: I could not understand collection procedures well. Was the flask flushed out prior to sample collection without dehumidifier before collection? I believe such kind of pre-rocess for flushing should be done with identical condition to sample collection. How long did you take for actual sample collection except for pre-flushing?*

Yes flasks were flushed out prior to sampling for ~10 minutes and flushing was done through the perchlorate (dehumidifier) column. The flasks were equipped with two stopcocks and after flushing the end stopcock was closed and allowed the pressure to build to 2 atm and then isolated by closing the other stopcock. This is discussed briefly in the revised manuscript (L 126-134). We also refer the details to our previously published papers such as Liang and Mahata (2015).

*L139: What is "systematic analyses"?*

"Systematic" refers to the study performed systematically, i.e., more regular and intensive sampling. To remove possible confusion, the word systematic is removed (L 138).

*L141-142; L146-147: Just to recommend, "..5 m high. It was closed at least one day before each experiment and the ventilation was kept as minimum as possible."*

Done (L 140)

*L150-155: Add the height of the canopy.*

Done (L 150)

*L155-157: Add each sampling height above sea level.*

Done (L 155-158)

*L169-174: If you used a vacuum line, add which process is in vacuo. If not, I'm sorry.*

Yes, we used vacuum line. $CO_2$ was extracted from air using a glass vacuum line connected to a turbo molecular pump by cryogenic technique. The vacuum line as well as the sample flask connection assembly including its head space was pumped to high vacuum before starting the $CO_2$ extraction. The details are mentioned in the revised manuscript (L 169-176). We also refer the details to our previously published papers such as Liang and Mahata (2015).

*L194-196: Specify the names of the standard (VPDB, VSMOW, etc., for each).*

Done (L 197)

*L212: What is "this limit"?*

The limit here refers to the full scrambling state. In this revised version, we replaced the term by "random distribution" (L 216)

*L217-219: Just to recommend, "Masses 48 and 49 were monitored to confirm isobaric interferences due to contamination of hydrocarbons (Ghosh..).*

Modified the sentence (L 221)

*L221-233: Refer Yoshida et al. (2013) RCM27, 207-215, for the evidence of independence from d47 on D47.*

Dependence of d47 on D47 varies from mass spectrometer to mass spectrometer. Therefore, this is not relevant, we discussed this in a previous publication (Laskar et al., 2016).

*L235, 237: I am not so sure whether this term is really appropriate or not, however "empirical transfer function" is often based on the field observation, such as marine foraminifer community structure versus habitat temperature. Authors obtained a relation experimentally, thus I think "reference frame equation", "laboratory equation" or "local equation" should be more appropriate instead of empirical transfer function.*

Though "empirical transfer function" is used by Dennis et al. (2011), we agree with the reviewer that the "reference frame equation" is more appropriate. This paragraph has been removed from the revised manuscript as it was discussed in another recent publication (Laskar et al., 2016).

*L237-239: Authors need not to discuss in detail, but should compare their results with former study.*

The reference frame equation varies between mass spectrometer to mass spectrometer, even it differs for a given mass spectrometer at different time. It is known to the community. This part is removed from the revised manuscript as it was discussed in a recent publication (Laskar et al., 2016).

*L245: The 1-sigma values of d13C and d18O seem too large whereas that of d47 seems in agreement with previous studies (Table S1). Huntington et al. (2009) described that d13C or*

*d18O uncertainties were roughly an order of magnitude better than d47, because of those higher abundance. Actually, Yoshida et al. (2013) showed these lower uncertainties accordingly. To my knowledge, in any way, if one measures d13C with [44] signal of 12V and integration time of 2.5 hour, the standard deviation may be better than 0.01 permil, not only for single gas but also for several aliquots. Actually, results of CO2 digested from carbonates (Table 1) are similar accordingly. Do you have any idea why uncertainties of cylinder CO2 became so high, or d47 uncertainty became lower relatively?*

For carbonates, it is possible to achieve a std. dev. of 0.01 (Table 1). For air CO2 (compressed cylinder air or atmospheric air), handling/purification worsen the precision. Though efforts have been put (see Liang and Mahata, 2015, for example), the best precision we can get so far for d13C and d18O is ~0.05 per mil. The precision we agree that is not sufficient for CO2 long term monitoring, but is sufficient for the current study. Possible cause is likely that slight fractionations during the extraction cause this variation in d13C and d18O. However, this possible fractionation does not impair the D47 analysis.

*L250-252: Add references for demonstrating poor consensus.*

Dennis et al., (2011), the inter-laboratory comparison shows D47 values of NBS-19 from 0.373 to 0.404‰ for three laboratories. This part is removed from the revised manuscript as it was discussed in a recent publication (Laskar et al., 2016)

*L254-255: Not only showing deviations from expected temperature, specify the reproducibility of D47 thermometry.*

This is discussed in a previous publication (Laskar et al., 2016) and removed from the revised manuscript

*L267-272: Lack of data source of temperature at South China Sea.*

Actual measurements during sample collection, mentioned in the revised manuscript (L 157).

*L276: Diurnal variation..*

Corrected (L 256)

*L282: Define Keeling plot and describe its purpose before the first use for readers from different fields.*

A brief description of Keeling plot and purpose is incorporated in the revised manuscript (L 261).

*L288: What is expected (potential) contamination of anthropogenic CO2 in the greenhouse?*

The potential contaminants are the ambient air with significant anthropogenic components which was found absent from [CO2] and all the isotope signatures.

*L296: What does "daytime" correspond? Daytime on 12th October? Or other three days?*
It is from morning 9 am to evening 5 pm, statement is modified in the revised manuscript (L 286).

*L297-299: The criteria of separation between weak/strong for photosynthesis or respiration in Fig. 4 is quite unclear. It seems very arbitrary. Define it clearly, otherwise delete this sentence and Fig. 4.*

By weak photosynthesis we wanted to mean that the photosynthetic activity was reduced artificially. This was done by covering the greenhouse with a double layered black cloth on a dark cloudy day. This is more clearly explained in the revised manuscript (276-294).

*3.2: Catalytic converter in the exhaust plays a role to convert CO to CO2. Is there any possibility this catalytic reaction may affect d18O value as same as D47, not only by exchanging oxygen with water?*

The change in d18O in the exhaust was also observed (Sec. 3.2). We are not aware of any process other than exchange of oxygen isotopes between CO2 and condensed water which can cause the change in the d18O or D47 of the exhaust CO2.

*3.3: This section should be divided into each field and reorganize to avoid confusion. For example, marine (including SCS and coastal sites), urban (Roosevelt Road), sub-urban (AS), grassland (NTU) and mountain. I guess authors might confuse a bit. For example, the relations between CO2 (1/CO2) and d18O as well as d13C and d18O for grassland are significant (regressions were done with data from Table 5), unlike its statement found in L346-349. Incidentally, the order to explain d13C and d18O results is marine, urban, sub-urban, grassland, mountain then forest. On the other hand, that to explain D47 results is marine, sub-urban, grassland, forest, mountain then urban. Easy to confuse.*

We reorganized the sections (Sec 3.2 to 3.4) and presentation is consistent in the revised manuscript. Section 3.4 is divided into several paragraphs to remove confusion.

*L314-328, L368-372: These should be reorganized as a separate section "marine CO2" for example.*

Marine and coastal CO2 data are presented in a new section (Sec 3.3)

*L330-333, L386-390: These should be reorganized as a separate section "urban CO2" for example.*

Urban, sub-urban, grass-land, forest and high mountain CO2 data are presented under one section (Sec 3.4) but separated into paragraphs.

*L333-339, L372-376: These should be reorganized as a separate section "sub-urban CO2" for example.*

Urban, sub-urban, grass-land, forest and high mountain CO2 data are presented under one section (Sec 3.4) but separated into paragraphs.

*L339-349, L376-379: These should be reorganized as a separate section "grassland CO2" for example.*

Urban, sub-urban, grass-land, forest and high mountain CO2 data are presented under one section (Sec 3.4) but separated into paragraphs.

*L353-357, L379-384: These should be reorganized as a separate section "forest CO2" for example.*

Urban, sub-urban, grass-land, forest and high mountain CO2 data are presented under one section (Sec 3.4) but separated into paragraphs.

*L314-328: The analysis based on the Keeling plot and subsequent source identification may be problematic. First, authors did not clarify whether the ocean of the study area/period is source or sink of CO2. Second, data range both of CO2 and d13C are narrow and number of data is limited, thus intercept of regression line must have large uncertainties. Therefore, some sentences from L324 to 328 and associated discussion in Section 4 may not be so meaningful. Moreover, authors should consider marine air interacts with ocean surface layer (mixed layer), not with deep ocean directly. The inconsistency between opaque Keeling intercept and d13C value from unconnected deep ocean is not surprising at all.*

We agree with the reviewer about the application of Keeling plot with a few data points covering a small range. The region is a net source of CO2 in the atmosphere, discussed in the revised manuscript (Sec 4.3). We put less emphasis on the Keeling plots over the ocean in the revised manuscript.

*L331: 39.32 instead of 39.319*

Done (L 328)

*L332-333: The average d18O value is not different significantly from that of grassland, thus this explanation is partly incorrect.*

The mean values are significantly different though the uncertainty associated with the values is large. d13C values are significantly different, but it is difficult to conclude based on d18O as mentioned in the later part of the section. The statements are modified in the revised manuscript (L 328).

*L344-345: I agree with this conclusion, however not by the result from Keeling plot, but by strong relations of CO2-d18O and d13C-d18O as mentioned above. D47 result may support this, thus I would like to emphasize that all results from same field should be described at once (in same block), should not be separated. However, this kind of concluding remark is supposed to be in the discussion.*

We agree that this should be discussed as a block in the discussion; this is moved to discussion (Sec. 4.5).

*L346-349: I totally disagree with this sentence. Authors should verify data again.*

Away from, for example, significant anthropogenic sources, due to presence of a variety of water sources (leaf water, soil water, etc), correlation between 1/[CO2] and d18O is always not observable.

*L358-367: This block and Fig. 6 may not be necessary.*

This paragraph along with Fig. 6 has been removed from the revised manuscript.

*4: The section and order of description is inconsistent with Results. This prevents readers from moving on smoothly. Consider above mentioned comment and reorganization.*
We thank the reviewer for the suggestion. This section is totally reorganized in the revised manuscript.

*L400-418: These blocks should move to introduction.*

Removed from the revised manuscript

*L422: "biological" instead of "biogeochemical"*

Done (L 376)

*L437-446: The obtained fractionation factor of -15.3, which is significantly different from expected C3-type fractionation, clearly demonstrated that this calculation is not applicable to the photosynthesis-respiration coexisting process. Authors should consider the different approaches. For example, assuming constant respiration rate for whole day (applying night time respiration rate to daytime), then obtaining gross productivity.*

We agree with the reviewer that the calculation should include respiration also. We modified our calculation assuming a constant respiration and presented the estimate in the revised manuscript (L 396).

*L446-454: Describe how consistent with previous studies. Consider same calculation mentioned above.*
Calculation is modified as per the suggestion and the calculated d13C and d18O discriminations have been compared with previous studies in the revised manuscript (L 401-409).

*L455-489: Authors demonstrated that d18O of respired CO2 is out of equilibrium with ambient temperature (water is supposed to have constant value, thus disequilibrium is due to temperature variation). If so, D47 of respired CO2 must be always out of equilibrium as well unless d13C is disequilibrium in a same manner (difficult to postulate due to the different fractionation process). However, authors mentioned that respired CO2 is in equilibrium with temperature because data in the early morning or night-time show close to equilibrium. This is a contradiction in principle. With keeping this contradiction, authors developed further discussion with respect to catalytic reaction. I cannot say whether the discussion is correct or not, however, I can say authors ignores a significant contradiction in the same block. Temperature change during night-time and cloudy (sun-shaded in addition) daytime were small whereas sunny days had wide range of temperature. Simply considering, larger magnitudes of disequilibrium during sunny daytime may be attributed this large temperature variation. Alternatively or additionally, authors had better consider that air temperature may*

*be different from body temperature inside leaves. Plants have homeostatic function with respect to temperature, a transpiration. CO2 is respired inside the leaf in partial isotope equilibrium with body temperature, not ambient temperature. I believe authors could develop much more deep and quantitative discussion with data shown in this study, before measuring clumped isotope of O2.*

We did not say that d18O of respired CO2 is out of equilibrium. We only showed that the respired CO2 is in thermodynamic equilibrium with the leaf and soil water using the obtained D47 values. We agree with the reviewer that the plant body temperature could be different from the air temperature but with progress of the day we expect change in the D47 values. As stated in the later part of this section, this needs to be tested at leaf level which we are planning and hopefully, the results will help to understand/model the effect of photosynthesis on the D47 values.

*L469: Remove "we believe"*

Done (L 429)

*L474: Yeung et al., 2015).*

Done (L 444)

*L490-498: This block should move to Summary.*

Done (L 665)

*L501-513: As mentioned above, I find it difficult to understand why authors would like to link atmospheric CO2 to respired CO2 in the deep ocean. I think this is unnecessary, and recommend to remove entire this block.*

We agree with the reviewer and reduced the discussion in the revised manuscript. However, we think that some explanation of the observed is required and kept a paragraph on this (L 494-507).

4.3: As mentioned above, authors had better consider the possibility of catalytic reaction between CO and CO2 at the converter.

Yes reaction between CO and CO2 inside the catalytic converter at the temperature of the converter could also lead to the change in the D47 values, though this would not change in the d18O values as the source of O2 in both CO and CO2 is the atmospheric O2. This is discussed in the revised manuscript (Sec. 4.2).

4.4: Authors gave f, anthropogenic contribution, in the two-source component equation from the difference between observed (urban) and marine CO2. This assumption ignores photosynthetic uptake or influence of other sources completely. Authors should get f by solving simultaneous equations based on the concentration and isotopic composition, conversely, then discuss. This approach may be more purposeful, quantitative and premised (why isotope study is needed).

We agree with the reviewer that a more quantitative estimate for CO2 cycling fluxes between reservoirs is possible. However we note that for example, atmospheric transport, that we mentioned at the end of the section, can easily interfere the calculation (box model interpretation, for example). This is the main reason that we give a more quantitative assessment for the greenhouse data, but not ambient CO2 data.

or new 4.6: A trial to estimate individual fluxes of combustion, respiration and photosynthesis for C3 and C4, respectively, from/to the urban (or sub-urban) CO2 is very welcome by using [CO2], d13C, d18O and D47.

Please see the previous response. We agree that the multiple CO2 isotopologues can help to constrain the CO2 fluxes of combustion, respiration and photosynthesis for C3 and C4, etc. However, incomplete knowledge on meteorological influence and lack of systematic dataset around the region prevent us from full assessment. From the available data presented, we showed that D47 behaves differently from [CO2], d13C, and d18O. To minimize regional and/or global interference (due to atmospheric transport, for example), we use greenhouse as a testbed for assessing the associated biological CO2 fluxes. For combustion, there are other tracers more useful than the presented CO2 isotopologues, such as VOCs and 14C.

*Fig. 1: Detail map of collection site in the Taipei city is desirable instead of right panel. Coastal and mountain sites can be involved into the left panel.*

Done (Fig 1)

*Fig. 3C: Although there appears a fair negative relation between d18O and D47 in Figs. 3A, B and D, coordinated rapid drops subsequent increases of these values are found on 4th August (3C) as well as 31st July. Do you have any idea what happened at these periods?*

Actually the correlation is significant only in Figure 3D. The reason for the rapid decrease in the D47 values in the early in response to photosynthesis is not very clear. We are doing more study at leaf level to identify the possible cause.

*Fig. 4: As mentioned above, the criteria to separate A and B is unclear.*

Here we wanted to show that D47 values are similar to that expected thermodynamically when respiration is strong and photosynthesis is weak but not the other way round. This is elaborated in the revised manuscript (Sec 4.1)

*Fig. 5: Data from urban site should be added. Ocean and coastal sites can be merged.*

Urban site data incorporated (Fig 7A). Ocean and coastal site merged in Fig 5.

*Fig. 6: This figure is unnecessary (see above).*

Removed from the revised manuscript

*Fig. 7: Reorganize (rearrange) according to the order of results and discussions.*

Done

*New Fig. 9?: The summarizing diagram for individual fluxes (schematic box diagram) is welcome.*

We agree that a summarizing diagram of individual flux will enhance the presentation. However, with the present data it will be too early to assign D47 values to individual fluxes. We will keep this suggestion in mind and try in future with more data.

*Table 2: Add relative humidity if available.*

We occasionally measured the relative humidity, not for all samples. We don't think that relative humidity can have major role in clumped isotopes.

**Anonymous Referee #2**
*The manuscript "Clumped isotopes in near surface atmospheric CO2 over land, coast and ocean in Taiwan and its vicinity" provided a valuable dataset of clumped isotopes in atmospheric CO2 and the authors did a good job. For the comments please see the attached file.*

We thank the reviewer for appreciating our effort. All the reviewers queries from the pdf and modifications/changes made in the revise manuscript are listed below. Also the other minor suggestions such as changing present/past tenses in the sentences, deleting/adding texts in the manuscript are be made in the revised manuscript.

*Line 28: The sentences should be in past tense.*

Done (L 29)

*Line 32: Not clear which processes. mention them i.e. photosynthesis, fossil fuel combustion ...*

The different processes are photosynthesis, respiration, local anthropogenic emissions, modified in the revised manuscript (L 32-33).

*Line 33: Split the sentence*

The sentence is modified (L 30-34)

*Line 34: Restructure the sentence: for example, the contribution of various sources of CO2 on D47 ...*

The sentence is restructured (L 34)
*Line 41: Split the sentence*

Done (L 40)

*Line 61: Split the sentence because it is hard to follow what you mean. Maybe: ... ocean and landbiosphere. The photosyn... 13C in plants is higher than ... .*

The sentence is divided into two for making it simple and easily understandable (L 61)

*Line 63: It is not clear what you mean. You should explain how photosynthesis and respiration may change 18O of CO2 in vicinity of plants, if it is what you wanted to say. Is there any discrimination against 18O during assimilation of CO2 for photosynthesis which may lead to enrichment or depletion in CO2 besides the leaves? In the next sentence your explanation just shows enrichement because of evapotranspiration but what is the effect of photosynthesis? Would be this isotopic discrimination due to evapotranspiration against 18O still present if the plant was not under water stress at all?*

The statements are modified as follows (L 68):
$\delta^{18}O$ is used for partitioning net $CO_2$ terrestrial fluxes between soil respiration and exchange with the plant leaves, the exchange is enhanced by the presence of carbonic anhydrase in plants and soils (Francey and Tans, 1987; Farquhar and Lioyd, 1993; Yakir and Wang, 1996;

Ciais et al., 1997; Peylin et al., 1999; Murayama et al., 2010; Welp et al., 2011). This is because $\delta^{18}O$ of $CO_2$ fluxes originated from soil respiration are different from that exchanged with the leaf water. $\delta^{18}O$ in soil water reflect the $\delta^{18}O$ value of the local meteoric water while leaf water is relatively enriched due to transpiration.

*Line 69: need reference*

Appropriate references are included (L 63)

*Line 71: This sentence should be in line 62 before 18O is used for partitioning ... .*

Done (L 66)

*Line 79: You mean reservoirs with different 18O?*

Yes, the statement is modified in the revised manuscript (L 78).

*Line 85: Split the sentence. You mixed many things together.*

Done (L 80)

*Line 86-96: Very well! This makes your study unique and valuable.*

We thank reviewer for appreciating the work

*Line 271: Materials and Methods is good.*

Thank you

*Line 277: The lowest CO2 concentration, [CO2] and the highest ...*

Corrected (L 257)

*Line 296: equilibrium with what? split the sentence.*
Thermodynamic equilibrium with the leaf and soil water, sentence modified and split (L 285).

*Line 300: my suggestion: The correlation between D47 and CO2... was observed only when the photosynthesis was weak.*

Suggestion implemented (L 291)

*Line 302: very good finding.*

Thank you

*Line 399: This paragraph can be deleted. It is not discussing any of the observation and measurements.*

This paragraph has been removed from the revised manuscript, in fact the first three paragraphs of the discussion are removed.

*Line 404: The sentences after "however" are not kind of discussion. I did not get why they should be mentioned here.*

These three paragraphs have been removed from the revised manuscript.

*Line 418: I think the whole these 3 paragraphs should be deleted. It is not clear what you wanted to say. Even if it was like an introduction for the discussion (which is not really necessary) you should follow to emphasis on the main issues respctively to what you will mention later for example effect of photosynthesis on D47, antropogenic effects in urban regions, ... .*

These three paragraphs have been removed from the revised manuscript.

*Line 477: Split the sentence*

Done ( L 352)

*Line 505: put the reference here and split the sentences.*

Done (L 496)

*Line 507: It is better to mention the intercept value here*

The intercept value is mentioned now (L 499)

*Line 511: write the value*

Done (L 505)

Line 521: refer to the fig. or value here

Done (L 516)

*Line 529: So D47 values in CO2 over oceans at nights should show no deviation from thermodynamic equilibrium. Is that true? How would be this effect in coastal areas where because of shallow water aquous plants may live as well?*

Yes, there should not be any deviation in the D47 in night also. The effect of photosynthesis on clumped isotopes is observable when photosynthesis is very strong e.g., in a confined greenhouse. Probably effect is present everywhere but not detectable with the measurement precision. Therefore, in the coastal areas we expect similar D47 values as observed over the open ocean unless there is a significant $CO_2$ is contributed from the other sources such vehicle and industrial emissions.

*Line 559: It seems logical but how? Do you have an estimation of isotopic composition of condensed water? How CO2 isotopic composition can change? I mean CO2 will dissolve in water but how its isotopic composition can change?*

Unfortunately we don't have any measurement of the d18O value of the condensed water but it is expected to be similar to the atmospheric O2 plus the fractionation associated with the condensation (atmospheric O2 is used for combustion). CO2 readily exchanges oxygen isotopes when comes in contact with water, here probably a partial exchange takes place causing the deviation from the expected d18O and D47 values. This section is more elaborately discussed (Sec 4.2).

*Line 562: Split the sentence! It is hard to follow you.*

This section is rewritten (Sec 4.2) in the revised manuscript

*Line 566: mention the temperature*

Temperature is mentioned (L 489)

*Line 583: reference needed*

Done (L 556)

*Line 589: split the sentence*

Done (L 564)

*Line 604: Can it be also less anthropogenic contribution?*

This value was obtained after subtracting the anthropogenic contribution. It can also be due to underestimation of the anthropogenic $CO_2$ at the sampling spot. The regional background $[CO_2]$ here could be lower than that assumed and the actual anthropogenic fraction of $CO_2$ could be higher than that assumed here. Discussed in the revised manuscript (L 576).

*Line 620: split the sentence.*

Done (L 597)

*Line 625: How could be anthropogenic effects in a dense and isolated forest area?*

It is very unlikely to have anthropogenic CO2 in an isolated place, but we did not neglect a priory. Later using D47 we showed that there is no anthropogenic CO2 in the forest site.

*Line 628: I think still you cannot exclude the photosynthesis effect. Furthermore, the humidity in such condition should be high which may modify the effect of respiration and may affect the temperature. You need to discuss about these issues.*

We agree that we cannot exclude photosynthetic effect. The issue is discussed more elaborately in the revised manuscript (L 625).

*Line 648: You need to discuss here about the photosynthetic effect on D47 values, especially that you took the samples just above the grasses.*

We agree with this suggestion and discussed more on this in the revised manuscript (L 620).

*Line 657: The fig. 8 is really a good job.*
Thank you

*Line 671:* explain it here also how and through which mechanism.

The explanation is also given here (L 660)

*Line 678: You did not discuss about this before. Delet this or explain the relation in the text. Why not "atmospheric CO2 budget" instead?*

This is removed from the revised manuscript.

**Anonymous Referee #3**

*This manuscript reports new measurements of clumped isotope compositions of atmospheric CO2 collected from different environments and settings. Studies of clumpedisotope composition of atmospheric CO2 were among the first applications of clumped isotope methods, but have received less attention in recent years compared to other applications. It's great to see another focused study on this subject. The dataset presented in this study is quite extensive, and mostly confirms the major findings from previous studies. However, the conclusion the authors draw regarding the effect of photosynthesis on the clumped isotope composition of CO2 differs significantly from previous studies, and could potentially open many research opportunities. Overall, this manuscript improves our understanding of the various controls on the clumped isotope composition of atmospheric CO2, and can help future efforts to better constrain the atmospheric CO2 budgets. I have several specific comments about this manuscript, as detailed below, and would recommend these issues be addressed prior to publication.*

We thank the reviewer for appreciating the work. Effect of photosynthesis on the clumped isotopes of atmospheric CO2 is an interesting finding and will be discussed more elaborately in the revised manuscript. We agree with all the concerns of the reviewer and addressed them in the revised manuscript.

Major comments:
*1. Separation of N2O from CO2. A GC column was used to separate N2O from CO2 in this study. The authors showed a reasonable separation of the two in Fig. S2, but didn't mention the exact CO2 trapping time in their experiments. It's possible the CO2 yield was compromised in order to achieve the optimal separation of N2O. The authors need to provide more details and discuss how the compromised yield and/or residual N2O might affect their clumped isotope data.*

We collect CO2 during 14 – 23 min. Yes CO2 yield was slightly compromised in order to achieve the optimal separation of N2O. The recovery of CO2 was always greater than 95 %. The residual N2O is checked comparing the sample and reference 48 and 49 signals. If value of D48 is large (>5) we did not consider the measurement, either we clean the sample again or just through. When the signal voltage for 49 between sample and reference is more than 0.3 mV we also neglected those numbers. We have discussed this in a recently published article (Laskar et al. Terra Nova, 2016) and cited at appropriate place.

*2. Photosynthesis effect. In their greenhouse experiments, the authors observed that the clumped isotope compositions of CO2 were higher than what expected from thermodynamic equilibrium when photosynthesis was active. This finding is very intriguing and differs from what observed in previous studies (e.g. Eiler and Schauble 2004), where the clumped isotope compositions of CO2 residual to photosynthesis were shown to generally decrease.*

Please see a previous response where the issue is discussed.

a. *Given the importance of this finding, I think the authors need to provide D48 and D49 data of their measurements to show that the elevated D47 values were not related to any contamination issues. More generally, the authors are encouraged to include all their raw clumped isotope measurement data in the electronic supplementary material of their manuscript, which is becoming a convention in the clumped isotope community.*

We mainly monitor the contamination with D48 signals. We will include the D48 values for all the sample. We have D49 values also but our experience is that it is also controlled by the 44 signals.

*b. The authors need to expand their discussion about the clumped isotope effects associated with photosynthesis they observed, especially in relation to the findings in Eiler and Schauble (2004), and explore ways to reconcile the findings from the two studies.*

Please see a previous response where the issue is discussed.

*c. The authors did a nice job estimating the carbon and oxygen isotope fractionations associated with photosynthesis in their greenhouse experiments. But their discussion about the clumped isotope effect is mostly qualitative. The authors might want to construct a simple (semi-)quantitative model to simulate the evolution of the concentration and isotopic composition of CO2 in their greenhouse experiments. Such a model might enable them to quantitatively estimate the clumped isotope effects associated with photosynthesis, which would be an important contribution of this study.*

Please see a previous response where the issue is discussed.

Minor comments:
*1. Line 440: the authors neglected the daytime respiration when estimating the isotope effects associated with photosynthesis. They need to provide evidence to support this approach.*

We agree that this is a crude assumptions, we have done the calculations incorporating the day time respiration in the revised manuscript (Sec 4.1).

*2. In section 4.1, the authors estimated the rates of respiration, photosynthesis, and CO2-water exchange in their greenhouse experiments, in the unit of molecules cm-2 s-1. But it's not entirely clear how those values were derived. More details are needed.*

We have provide more details about the calculations in the revised manuscript (L 401).

[revised manuscript text omitted]

Inserted Cells

Inserted Cells

| 08/18/2015 10:30 | 433 | -9.99 | 39.80 | 8.99 | 0.02 | 0.888 | 0.016 | 0.4 | 32 | 30 |
|---|---|---|---|---|---|---|---|---|---|---|
| Average ± 1σ | 438±16 | -9.99 ±0.50 | 40.39±0.66 | 9.82±0.94 | | 0.895±0.012 | | | 31±2 | 31±1 |
| **High mountain: Hehuan** | | | | | | | | | | |
| 10/09/2013 13:20 | 364 | -8.21 | 40.89 | 28.79 | 0.02 | 0.895 | 0.016 | 3.2 | 31 | 10 |
| 10/09/2013 17:00 | NA | -8.25 | 40.28 | 28.41 | 0.01 | 0.914 | 0.014 | 2.9 | 27 | 10 |
| Average ± 1σ | 364 | -8.23 ±0.02 | 40.59±0.30 | 28.60±0.19 | | 0.904±0.009 | | | 30±2 | 10 |

Inserted Cells